



# Modelling surface temperature and radiation budget of snow-covered complex terrain

Alvaro Robledano[1,2], Ghislain Picard[1], Laurent Arnaud[1], Fanny Larue[1], and Inès Ollivier[1]

[1]Univ. Grenoble Alpes, CNRS, Institut des Géosciences de l'Environnement (IGE), UMR 5001, Grenoble, F-38041, France
[2]Univ. Grenoble Alpes, Université de Toulouse, Météo-France, CNRS, CNRM, Centre d'Etudes de la Neige, 38000 Grenoble, France

**Correspondence:** Alvaro Robledano (alvaro.robledano-perez@univ-grenoble-alpes.fr)

**Abstract.** The surface temperature controls the temporal evolution of the snowpack playing a key role in many physical processes such as metamorphism, snowmelt, etc. It shows large spatial variations in mountainous areas because the surface energy budget is affected by specific radiative processes that occur due to the topography, such as the modulation of the irradiance by the local slope, the shadows and the re-illumination of the surface from surrounding slopes. These topographic effects are often neglected in large scale models considering the surface as flat and smooth. Here we aim at estimating the surface temperature and the energy budget of snow-covered complex terrains, in order to evaluate the relative importance of the different processes that control the spatial variations. For this, a modelling chain is implemented to derive surface temperature in a kilometre-wide area from local radiometric and meteorological measurements at a single station. The main component is the Rough Surface Ray-Tracing model (RSRT), based on a photon transport Monte Carlo algorithm to quantify the incident and reflected radiation on every facet of a mesh, describing the snow-covered surface. RSRT is coupled to a surface scheme in order to estimate the complete energy budget from which the surface temperature is solved. To assess the modelling chain performance, we use in situ measurements of surface temperature and satellite thermal observations (TIRS sensor aboard Landsat-8) in the Col du Lautaret area, in the French Alps. The satellite images are corrected from atmospheric effects with a single-channel algorithm. The results of the simulations show (i) an agreement between the simulated and observed surface temperature at the station for a diurnal cycle in winter within $0.3\,°C$; (ii) the spatial variations of surface temperature are on the order of 5 to $10\,°C$ between opposed slope orientations and are well represented by the model; (iii) the importance of the considered topographic effects is up to $1\,°C$, the most important being the modulation of solar irradiance by the topography, followed by altitudinal variations in air temperature, long-wave thermal emission from surrounding terrain, spectral dependence of snow albedo, and absorption enhancement due to multiple bounces of photons in steep terrain. These results show the necessity of considering the topography to correctly assess the energy budget and the surface temperature of snow-covered complex terrain.



# 1 Introduction

The snow surface is rarely flat and smooth on Earth. Undulations exist over a very large range of scales. At the centimetre and metre scales, ripples, snow dunes, and erosion features (sastrugi) formed by wind usually coexist (Filhol and Sturm, 2015).

Penitents (spike formations of snow and ice – Lliboutry (1954)) are also found in some particular conditions. At the decametre to kilometre scale range, the snow surface topography is mostly determined by the underlying soil or ice topography (Revuelto et al., 2018). Because of all these undulations, the surface temperature can vary by several Celsius degrees across a study area, even without significant differences in the near-surface meteorological forcing (wind, air temperature, humidity). Terrain tilt and the presence of facing neighbouring terrains cause significant variations in the surface energy budget and more specifically

in the radiative components which comprise the short-wave (SW) and long-wave (LW) radiative fluxes. An abundant literature has investigated how SW and LW radiation is distributed across a complex terrain (Marks and Dozier, 1979; Duguay, 1993; Plüss and Ohmura, 1997), often for applications in mountainous areas. Nevertheless, even if the literature for the smaller scales – that of the ripples, dunes, sastrugi and penitents – is usually distinct and scarcer, the principles equally apply to all the scales because the radiative transfers between faces are invariant by scale change. The exception is over long distances when the

atmospheric scattering and absorption effects due to the air present between the terrain faces become significant (Lamare et al., 2020).

This study investigates and quantifies the relative importance of a series of topographic effects that control the radiative budget and the surface temperature. The first effect, applying during daytime and under clear sky conditions, is the modulation of the solar irradiance received by a face depending on its slope and aspect relative to the sun's position. This modulation

depends on the cosine of the local solar zenith angle. Self-shadowing occurs when the face completely turns away from the sun, that is when the local solar zenith angle is below 0° or over 90°. Chen et al. (2013) accounted for this first effect, all the other terms of the energy budget being calculated as for flat terrain. This approximation is called "small slope approximation" by Picard et al. (2020) which estimate it to be valid for gentle slopes up to $\approx 20°$.

Arnold et al. (2006) investigated the topographic parameters that control the surface energy balance on an Arctic glacier.

Their model takes into account not only the local zenith angle and cast shadows for direct SW radiation, but also the sky view factor for the diffuse SW and sky LW radiation. The model additionally accounts for LW radiation from the surrounding slopes by assuming an average surface temperature, a simplification with respect to the reality where each face may have a different temperature. The model evaluated with measured melt rate in different parts of the glacier performs well (correlation coefficient $r \approx 0.85$ for the majority of the melt season). The topographic effects are ranked by order of importance in determining the

surface energy budget, first shadowing, second the local zenith angle and third the sky view factor. Olson et al. (2019) confirmed these findings regarding the two first effects. Arnold et al. (2006) also pointed out the role of the anisotropic reflectance of snow and ice, i.e. the fact that albedo is higher at higher solar zenith angles (Warren and Wiscombe, 1980).

Absorption enhancement is an additional effect, particularly important in steep terrains. It is indeed likely that the incident photons in such a terrain encounter more than one bounce between the terrain faces before escaping to the sky or being

absorbed in the snow. The total probability of absorption of a given incident photon is thereby increased by the successive





bounce compared to over a flat or smooth surface where only at most one bounce occurs (Warren et al., 1998). Larue et al.
(2020) experimentally quantified this effect by measuring spectral albedo over artificial rough surfaces and nearby smooth
surfaces, effectively showing a decrease of albedo (i.e. increase of absorption) of the order of 0.03 – 0.08 in the visible and
near-infrared. This range was confirmed using optical SW-only models for rough surfaces (Warren et al., 1998; Leroux and
Fily, 1998; Lhermitte et al., 2014). For real 3D terrains, ray tracing and Monte Carlo techniques can take into account this
effect (Larue et al., 2020), at the expense of intensive computation. A simpler approach to account for multiple bounces is
by assuming that the neighbouring faces are illuminated as if they were flat (Lenot et al., 2009; Olson et al., 2019). More
importantly, the absorption enhancement is not uniform on the surface. It is usually stronger in the valleys (trapping effect or
focusing) resulting in a higher ablation (melt or sublimation) observed in the valleys compared to that near the summits of
the topography (Lliboutry, 1954). When the scale of the surface roughness is of the order of the ablation rate (e.g. crevasses,
sastrugi, penitents, ...), this differential ablation can significantly affect the topography, and in fact can increase the amplitude
of the terrain undulations, and thus further increase absorption. This positive feedback loop is key to explain the formation
of penitents (Cathles et al., 2014) (and refs therein) and suncups (Betterton, 2001). The energy balance model developed by
Cathles et al. (2011) and Cathles et al. (2014) to explore this feedback considers 2D roughness (e.g. a linear crevasse) and
computes absorption in every small element of the surface. While their model includes shadowing, local zenith angle and
multiple bounces, it neglects the view factor on sky LW radiation and emission by the faces, which could further enhance
absorption in the valleys. Corripio and Purves (2005) also investigated penitent formation, and accounts for LW radiation and
sky view factor effect with a similar approach to that of Arnold et al. (2006) for mountains. LW emission by faces can be a
significant term in a steep topography, as in penitents. At larger scales (kilometre), Yan et al. (2016) showed that neglecting
topography effects may induce errors of up to $100\,\mathrm{W\,m^{-2}}$ for the modelled LW net flux. Lee et al. (2013) used 3-D Monte
Carlo photon tracking methods to show the impact on surface radiation budget over the Tibetan Plateau, finding deviations in
surface solar fluxes on the order of $14\,\mathrm{W\,m^{-2}}$.

The turbulent heat fluxes are of significant importance in the surface energy budget, as shown by Brun et al. (1989). They
are capital in snow-covered areas, and in particular during nights, as they balance the radiative cooling of the surface. These
fluxes are difficult to parameterized as only a few in situ measurements are usually available, and only at a single level (Martin
and Lejeune, 1998; Pomeroy et al., 2016). In complex terrain, their spatial and temporal variations are mostly determined by
the wind distribution around the relief and by change of air temperature with elevation (Rotach and Zardi, 2007).

The most advanced model to simulate surface temperature in a mountainous area is – to our knowledge – a commercial
product (Adams et al., 2009, 2011) which includes not only a comprehensive energy budget modelling (inc. absorption en-
hancement and emission by faces) but also radiation penetration in the snow, different types of surface (snow, rock, tree) and
the heat diffusion in the snow to describe the full temperature profile in the snowpack for each face. The precise equations and
approximations are however not known. In a study case in southwest Montana, USA, the model predicts spatial variations of
temperature between -15 °C and 0 °C at 2 pm local time, with most of these variations being primarily explained by terrain
type. This non-exhaustive review of approximations and models in the literature highlights the various effects related to the



topography, the large number of possible approximations and the degrees of complexity used at both metre-scale roughness and complex-terrain scales. These effects and approximations are summarized in Table 1 and illustrated in Fig. 1.

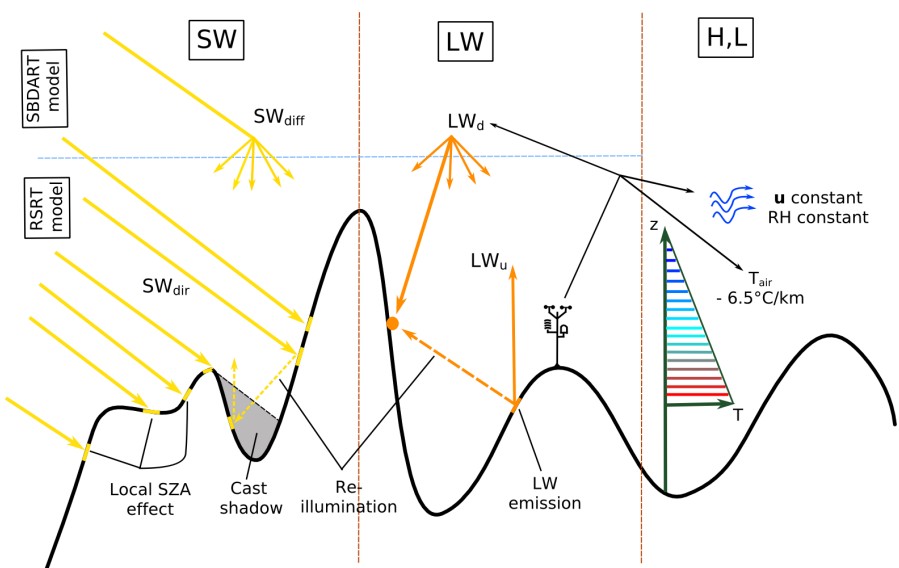

**Figure 1.** Illustration of the several topographic effects that are considered in this study. Common approximations for the calculation of these effects are summarized in Table 1.

In addition, most models cited above consider only broadband SW fluxes, neglecting the spectral dependence of snow albedo and incident radiation. The probable consequence is an inaccurate estimation of the absorption enhancement due to neglecting the large difference of albedo between visible and near-infrared. In the former domain, the albedo is high (typically over 0.95),

implying intense multiple scattering but extremely weak absorption. In contrast, in the near-infrared the albedo is lower and closer to the optimal albedo for enhancement (0.5), where multiple scattering and absorption are balanced (Warren et al., 1998).

To investigate the relative importance of the topographic effects, this study aims at estimating the snow surface temperature in a mountainous area with a modelling chain that uses local in situ radiometric and meteorological measurements from a single

station. The chain includes a 3D radiative transfer model, the Rough Surface Ray Tracer (RSRT) model (Larue et al., 2020), which launches a set of photons to the snow surface described by a triangular mesh, i.e. a connected set of triangular facets, that can be derived easily from a Digital Elevation Model (DEM). Its spatial resolution limits the scope of this study to topography at the decametre to kilometre scale range. The simulation results are used in a surface scheme (RoughSEB model) to compute the short-wave and long-wave net radiation and the turbulent heat fluxes on each facet. Eventually the surface temperature is

deduced. Satellite thermal infrared observations from Landsat-8 are used in order to evaluate the modelled spatial variations.



**Table 1.** Several effects relevant to short-wave (SW), long-wave (LW) and turbulent heat fluxes calculation in complex terrains and rough surfaces. Other effects such as those involving the atmosphere are beyond the scope of this study.

| Effect | Spectral domain and illumination |
|---|---|
| Variations of illumination angles and self shadows | Direct SW |
| Cast shadows | Direct SW |
| Anisotropy of reflectance | SW |
| Spectral variations in albedo and irradiance | SW |
| Face re-illumination / absorption enhancement | SW |
| Reduced sky view | Diffuse SW and LW |
| Face thermal emission | LW |
| Altitudinal changes in air temperature | Turbulent heat fluxes (H, L) |

To quantify the relative importance of the topographic effects, additional simulations are run disabling one single effect at a time and measuring the impact on the surface temperature. This study is applied at the Col du Lautaret area, in the French Alps. Sect. 2 provides the model description, as well as the method for retrieving surface temperature from satellite images and a description of the study area. Results are shown in Sect. 3, and discussed in Sect. 4. Final remarks and conclusion are addressed in Sect. 5.

## 2 Methods and materials

### 2.1 The RoughSEB model

Snow surface temperature ($T_s$) is obtained by solving the surface energy budget equation. Each term of this equation is estimated with a chain of existing and new models depicted in Fig. 2. The energy budget comprises (Arya, 1988): (i) the net radiation fluxes, which are split into the contributions of the short-wave radiation from $0.3\,\mu m$ to $2\,\mu m$ ($SW_{net}$) and the long-wave radiation from $2\,\mu m$ to $100\,\mu m$ ($LW_{net}$); (ii) the sensible heat flux ($H$), which measures the exchange of heat between the surface and the air just above; (iii) the latent heat flux ($L$) resulting from water state changes (sublimation or condensation) at the surface and exchange with the air above, and (iv) the ground heat flux ($G$), which is transferred to the snowpack and lead to a change of temperature or melt. Here, we consider only temperature estimation and assume below freezing temperature so that melt is not involved. The sum of these fluxes is null according to the first principle of the thermodynamics, taking into





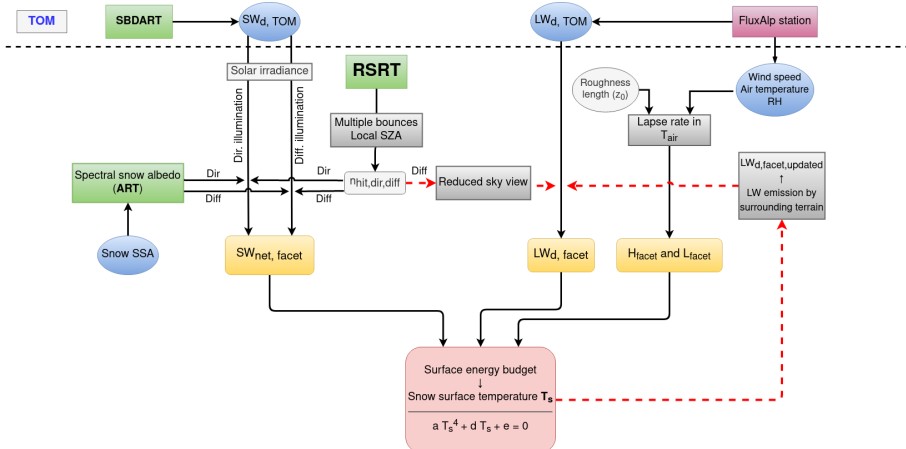

**Figure 2.** Flowchart of the modelling chain to estimate snow surface temperature. The models involved are in green, and the terms of the surface energy budget are in orange. The red dashed lines indicate the last steps of the chain.

account that the surface has no internal enthalpy:

$$SW_\text{net} + LW_\text{net} + H + L + G = 0 \tag{1}$$

In the RoughSEB model, all these terms are calculated for each facet (hereafter noted with the subscript f) of the modelled snow surface. The computation of the $SW_\text{net}$ radiation (Sect. 2.1.1) involves: (i) the Santa Barbara DISORT Atmospheric

Radiative Transfer model (SBDART – Ricchiazzi et al. (1998)), an atmospheric model that simulates the incoming spectral irradiance above the studied area (top of mountains – TOM), and (ii) the RSRT model (Larue et al., 2020), based on a Monte Carlo photon tracking algorithm that computes the path of the photons launched towards the surface and allows considering the modulation of SW radiation by the terrain slope and aspect. The simulations are run in both direct and diffuse illumination conditions (noted with subscripts dir and diff), and the atmospheric effects (i.e. atmospheric attenuation) are neglected within

the studied area (between the surface and TOM). The scene is also considered to be covered by pristine, pure snow (i.e. no impurities), which is applicable in winter. The calculation of the $LW_\text{net}$ radiation (Sect. 2.1.2) needs the downwelling LW flux, measured by a local station representative of the area, and subsequently updated by accounting for the reduced sky-view factor and the thermal emission of surrounding terrain as in Arnold et al. (2006). The effect of the turbulent heat fluxes ($H$ and $L$), while being considered, is not the main objective of this work. They are computed with a simplified approach (Sect. 2.1.3) that

involves common in situ meteorological measurements (i.e. air temperature, wind speed) and introduces a lapse rate effect.

The snow surface temperature (Sect. 2.1.4) is eventually estimated for each facet of the modelled surface. The topographic effects that are taken into account in this study can be disabled or modified when needed, allowing to quantify their relative importance.





### 2.1.1 Calculation of the short-wave radiation fluxes

In order to consider the topographic effects affecting the SW radiation and the spectral dependence of snow albedo and incident radiation, the calculation of the net SW radiation needs to be carefully performed. The aim is to compute it on every facet $f$ due to the direct and diffuse irradiation:

$$SW_{\text{net, f}}(\theta_s) = SW_{\text{net, dir, f}}(\theta_s) + SW_{\text{net, diff, f}} \tag{2}$$

$$SW_{\text{net, dir, f}}(\theta_s) = \int_{\lambda_0}^{\lambda_1} A_{\text{dir, f}}(\lambda, \theta_s)\, I_{\text{dir}}(\lambda)\, d\lambda \tag{3}$$

$$SW_{\text{net, diff, f}} = \int_{\lambda_0}^{\lambda_1} A_{\text{diff, f}}(\lambda)\, I_{\text{diff}}(\lambda)\, d\lambda \tag{4}$$

where the direct and diffuse absorbed SW radiation by every facet are computed using an absorption coefficient $A_{\text{dir, diff, f}}$, derived from the RSRT model, and the spectral irradiance coming from the sky ($I_{\text{dir, diff}}(\lambda)$), issued from the SBDART model. The integration limits ($\lambda_0$ and $\lambda_1$) are respectively equal to 300 nm and 2000 nm.

A naive way to compute these terms would be to use the ray-tracing RSRT model to trace many photons for every wavelength
in the solar spectrum to compute the absorbed SW radiation by each facet. Nevertheless, with millions of facets and hundreds of wavelengths, this approach would imply an enormous computational cost, due to the inefficiency of the Monte Carlo ray tracing method. To overcome this, an alternative approach consists of taking advantage of the result of the RSRT model to account for multiple bouncing provided a few assumptions. The RSRT model can indeed compute the number of times a photon has hit a given facet regardless of the albedo (and so of the wavelength), according to the bounce order of the photon
(first reflection, second reflection, ...). Noted $n_{\text{hit, d, f}}^{(i)}$, it corresponds to the proportion of photons that hit the facet on their $i^{\text{th}}$ hit (d being dir or diff depending on the illumination conditions – direct or diffuse).

Assuming the area has an uniform albedo (same snow properties) and the reflection is Lambertian, the absorption coefficient is computed by taking into account (i) the spectral dependence of snow albedo, and (ii) the fact that the illumination received by each facet depends on the cosine of the local solar zenith angle and the absorption enhancement produced by multiple
bouncing, with:

$$A_{\text{dir, f}}(\lambda, \theta_s) = (1 - \alpha_{\text{dir}}(\lambda, \theta_s))\, n_{\text{hit, dir, f}}^{(0)} + \sum_{i=1}^{i=n_{\max}} \alpha_{\text{diff}}^i(\lambda)\, n_{\text{hit, dir, f}}^{(i)} \tag{5}$$

$$A_{\text{diff, f}}(\lambda) = (1 - \alpha_{\text{diff}}(\lambda)) \sum_{i=0}^{i=n_{\max}} \alpha_{\text{diff}}^i(\lambda)\, n_{\text{hit, diff, f}}^{(i)} \tag{6}$$

where $\theta_s$ is the local solar zenith angle and $\alpha_{\text{dir, diff}}(\lambda, \theta_s)$ is the snow spectral albedo in both direct and diffuse illumination. Here, it is computed using the Asymptotic Radiative Transfer (ART) theory (Kokhanovsky and Zege, 2004). Its expression





is presented in the Appendix A1, provided several assumptions about the snowpack (semi-infinite, vertical and horizontal homogeneous layers) and the surface (flat and smooth – facets are small enough to be considered as it). It considers the snow specific surface area (SSA), which is needed as input in the modelling chain.

The net broadband SW radiation per facet is therefore calculated by means of Eqs. (2, 5, 6), previously accounting for the spectral irradiance coming from the sky and integrating each component over the 300 - 2000 nm wavelength range.

### 2.1.2 Long-wave radiation fluxes

The downwelling LW flux at the top of the mountains (noted $LW_{d, TOM}$) is issued from in situ measurements (Sect. 2.3) and considered constant across the scene regardless of the variations of altitude. To compute the downwelling LW flux incident on each facet, we proceed in two steps. In the first step, we consider the surface as flat so that all the facets receive the TOM LW radiation and no emission from the surrounding slopes. This leads to a first estimate of the surface temperature (Sect. 2.1.4) that is then used in the second step to account for the emission of surrounding slopes by computing the average upwelling long-wave from each facet as in Arnold et al. (2006), and using:

$$LW_{d, f, updated} = V_f \, LW_{d, TOM} + (1 - V_f) \, LW_{u, scene\text{-}average} \tag{7}$$

where $V_f$ is the sky-view factor estimated with RSRT. $V_f$ is different for each facet so that facets in the valley receive more energy from the surrounding slopes than facets at the summits of the mountains. It is indeed equal to the proportion of the launched photons hitting a facet on their first bounce in diffuse illumination, namely $V_f = n_{hit, diff, f}^{(0)}$. However, $LW_{u, scene\text{-}average}$ is a constant, we neglect the possible variations of temperature around each facet.

For the upwelling long-wave radiation, $LW_{u, f}$ is determined by the Stefan-Boltzmann law:

$$LW_{u, f} = \epsilon \, \sigma \, T_s^4 + (1 - \epsilon) \, LW_{d, f} \tag{8}$$

with $\epsilon = 0.98$, $\sigma$ the Stephan-Boltzmann constant and $T_s$ the snow surface temperature.

### 2.1.3 Turbulent heat fluxes

While the main focus of this work is on the role of the topographic effects controlling the radiative budget of the surface, the turbulent heat fluxes needs also to be assessed to compute the energy budget. We follow a very simple modelling approach at this stage, potential improvements are let to further work in the future. The sensible and latent heat fluxes are calculated following the one-level approach:

$$H_f = \rho_{air} \, c_{p, air} \, C_H \, U \, (T_s - T_{air}) \tag{9}$$

$$L_f = L_s \, \rho_{air} \, C_H \, U \cdot (q_{sat}(T_s, P_s) - q_{air}) \tag{10}$$

where $\rho_{air}$ and $c_{p,air}$ are the density and heat capacity of the air, $U$ is the wind speed, $T_{air}$ is the air temperature, $L_s$ is the sublimation heat, $q_{sat}(T_s, P_s)$ is the specific humidity at snow surface temperature $T_s$ and pressure $P_s$, and $C_H$ is a surface





exchange coefficient. In principle this coefficient depends on atmospheric stability but for the sake of simplicity, a neutral
situation is considered here. $C_H$ therefore depends only on the aerodynamic roughness length $z_0$, assumed constant across the
scene and equal to $10^{-3}$ m following previous works (Brock et al., 2006). The expression of this coefficient is provided in the
Appendix A2, and the values of the symbols defined here are presented in the Table B1. The air temperature and wind speed
are given from a meteorological station in the scene (Sect. 2.3). To account for the differences in altitude within the scene, the
lapse rate $\Gamma$ is taken into account:

$$T_{\text{air, f}} = T_{\text{air, obs}} + \Gamma \left( z_{\text{f}} - z_{\text{obs}} \right) \tag{11}$$

where $T_{\text{air, f}}$ is the air temperature over the facet $f$. We choose $\Gamma$ = -6.5 °C km$^{-1}$, the environmental lapse rate as defined in
the International Standard Atmosphere (ISO 2533:1975). Wind speed and relative humidity are however considered constant.
Note that the specific humidity being deduced from the relative humidity and air temperature, it indirectly depends on altitude.

The ground heat flux ($G$) is here neglected as the snowpack is considered thermalized, meaning that no energy is exchanged
with the snowpack. This assumption is checked in the discussion (Sect. 4.2) with regard to our results.

### 2.1.4 Snow surface temperature estimation

The equation of the surface energy budget (Eq. 1) is to be solved for $T_s$ for each facet $f$. However this equation is non-linear
due to the $T_s^4$ term (upwelling LW radiation) and the complex dependence of the surface specific humidity to $T_s$. Solving such
an equation for millions of facets can be computationally intensive. To avoid this issue, we linearize the specific humidity at
saturation:

$$q_{\text{sat}}(T_{\text{s}}, P_{\text{s}}) = q_{\text{sat}}(T_{\text{air}}, P_{\text{s}}) + (T_{\text{s}} - T_{\text{air}}) \cdot q'_{\text{sat}}(T_{\text{air}}, P_{\text{s}}) \tag{12}$$

where:

$$q'_{\text{sat}}(T_{\text{air}}, P_{\text{s}}) = \left( \frac{q_{\text{sat}}(T_{\text{air}}, P_{\text{s}}) - q_{\text{sat}}(T_{\text{air}} - \Delta T, P_{\text{s}})}{\Delta T} \right) \tag{13}$$

with a $\Delta T$ = 5 K. The simplified surface energy budget equation eventually become a quartic equation for $T_s$ of the form:

$$a\, T_{\text{s}}^4 + d\, T_{\text{s}} + e = 0 \tag{14}$$

where $a, d, e$ are a constant (for each facet). The general solution is shown in the Appendix A3 for completeness. The evaluation
of this equation for a large number of facets is computationally very efficient compared to using a non-linear optimization
technique.

## 2.2 Description of the existing models

The modelling chain involves two existing models, namely the SBDART model (Ricchiazzi et al., 1998) and the RSRT model
(Larue et al., 2020). Here we briefly describe the parameters to run them in the present work.





The SBDART model provides the short-wave irradiance spectrum at the top of the mountains. It considers several atmospheric transmission models and the Mie theory to take scattering into account. The simulations are run in the spectral range $300 - 2000\,\mathrm{nm}$, by steps of $3\,\mathrm{nm}$. In our case, a generic mid-latitude winter atmospheric profile is assumed and the aerosol
optical depth at $550\,\mathrm{nm}$ is 0.08 ("rural" profile in SBDART). The elevation is 2052 m.a.s.l. (see Sect. 2.3) and solar zenith and azimuth angles corresponding to the desired date and time.

RSRT model provides the number of photon hits on each facet according to its bounce order, noted $n^{(i)}_{\mathrm{hit,\,d,\,f}}$. In this model, each launched photon has an initial propagation direction $i$, specified by the zenith and azimuth angles, and a random initial position over the mesh. Edge effects are avoided by excluding the outermost 15% of the mesh. The model calculates the propagation
with the following main steps: (1) estimation of the next intersection between the photon path and the mesh; (2) determination of the reflection direction (random cosine-weighted hemispherical distribution in our case to simulate Lambertian reflection), and (3) update of the direction $i$. The algorithm iterates until the photon escapes from the scene or up to a maximum number of bounces of 15. The number of facets in the mesh is typically $10^6$ and the number of photons launched is $4 \cdot 10^8$.

## 2.3 Study area and in situ measurements

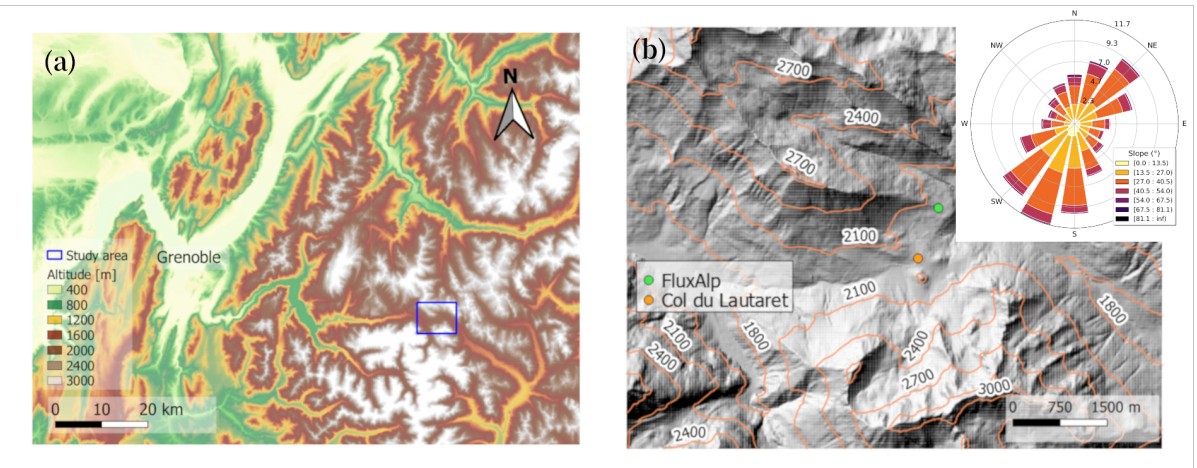

**Figure 3.** Location of the study area, around the Col du Lautaret alpine site. The blue rectangle in (a) represents the hillshade image shown in (b). It is generated from the RGE ALTI®Version 2.0 Digital Elevation Model (DEM) provided by IGN France with a spatial resolution of $5\,\mathrm{m}$, and resampled to $10\,\mathrm{m}$ for computational limitations. The windrose shows the distribution of the terrain slopes as a function of the aspect. The radius of each of the 16 sectors corresponds to the normed (displayed in percent) quantity of facets.

Figure 3 shows the study area. It is located around the Col du Lautaret in the French Alps (45.0°N, 6.4°E). This area is interesting for studying surface temperature, as it features both north and south-facing slopes, in addition to smaller-scale rugged terrain covering the rest of orientations and promoting re-illumination. The size of the area is $\approx 50\,\mathrm{km}^2$, with a large diversity of rough features. The predominant orientation is S-SW, followed by N-NE facing terrain, and the slope varies mainly between





15° and 40°. The mesh required for simulations was built from the RGE ALTI®Version 2.0 Digital Elevation Model (DEM)
provided by IGN France (IGN, 2021), acquired using radar techniques in 2009. Its coordinate reference system is Lambert 93
(EPSG: 2154), and the original spatial resolution was of $5\,\mathrm{m}$. It was however resampled to $10\,\mathrm{m}$ due to computational limita-
tions. To create the mesh, the centre of each pixel is a vertex that is connected to its closest neighbours, which eventually leads
to two triangular facets shared by the same vertex.

The study area also includes the measurement station FluxAlp (45.0413°N, 6.4106°E), on the Pré des Charmasses site.
This station collects meteorological and radiometric observations, and its description can be found in Dumont et al. (2017).
Radiation fluxes are measured with a Kipp & Zonen CNR4 radiometer. The data needed as input for the modelling chain is
issued from here (meteorological – air temperature, wind speed, relative humidity and radiometric – downwelling long-wave
radiation flux). The latter is assumed to be equal to that received above the area (TOM), as the LW re-illumination at the
station is very low according to initial tests of the modelling chain. FluxAlp measurements are averaged over $30\,\mathrm{min}$, i.e.
a measurement at 10h30 UTC is the average of the period 10h00 UTC – 10h30 UTC. The upwelling LW measurements at
FluxAlp are used to assess the modelled surface temperature temporal variations in a single point. They are complemented by
a spatially distributed surface temperature dataset described in the following section.

In addition to the automatic measurements, manual measurements of SSA for two consecutive winter seasons (2016 / 2017
and 2017 / 2018) have been collected occasionally (Tuzet et al., 2020). The measurements were collected with the DUFISSS
instrument (Gallet et al., 2009) during the first season, and with the Alpine Snowpack Specific Surface Area Profiler (ASSSAP,
a lighter version of POSSSUM instrument (Arnaud et al., 2011) during the second season. These measurements have an
estimated uncertainty of 10 %.

## 2.4 Surface temperature retrieval with Landsat-8 observations

Spatial variations of surface temperature are retrieved from satellite observations. The two thermal bands (TIRS – Bands 10
and 11) onboard Landsat-8 cover the spectrum between $10.6\,\mathrm{\mu m}$ to $12.51\,\mathrm{\mu m}$, with a spatial resolution of $100\,\mathrm{m}$ (resampled
by Cubic Convolution methods to $30\,\mathrm{m}$) and a 16 day repeat cycle. Different methods to correct atmospheric effects have
been implemented to retrieve Land Surface Temperature (LST, hereafter), based on split-window methods (Jin et al., 2015),
mono-window techniques (Tardy et al., 2016), or a single-channel approach (Jiménez-Muñoz and Sobrino, 2003). Soon after
the launch of Landsat-8, stray light was observed on thermal data (Montanaro et al., 2014), coming from scattering of outer
radiance in Band 11. Methods based on only one band are therefore suggested, so here we apply a single-channel approach,
which consists of approximating the atmospheric functions from atmospheric water vapour content ($w$, in $\mathrm{g\,cm^{-2}}$). Cristóbal
et al. (2018) presented an improved single-channel method dependent not only on water vapour content, but also on near-
surface air temperature ($T_\mathrm{a}$), which are available from reanalysis data. The recently available Landsat Collection 2 Surface
Temperature product is also considered here and compared to the results of the aforementioned LST retrieval methods in our
particular case of study. Figure 4 shows the flowchart to retrieve LST from satellite observations, following Cristóbal et al.
(2018).



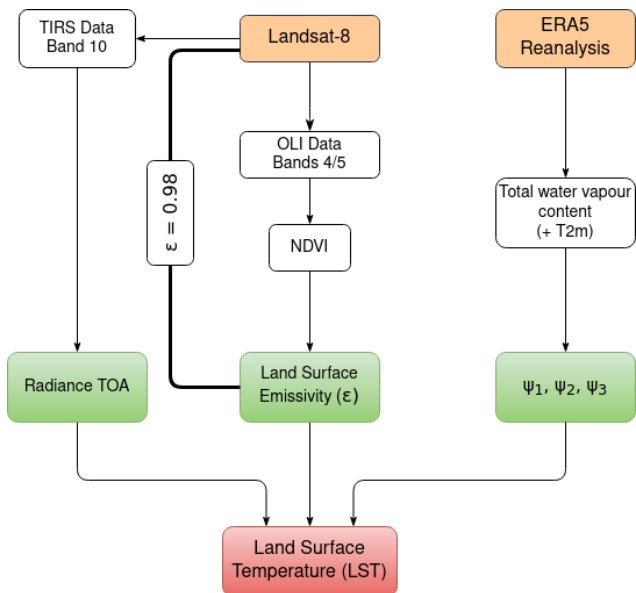

**Figure 4.** Workflow to retrieve Land Surface Temperature from Landsat-8 thermal observations with a single-channel approach.

Both the single-channel method (SC method – Jiménez-Muñoz and Sobrino (2003)) and the improved single-channel method (iSC method – Cristóbal et al. (2018)) have been implemented in this study. LST is calculated for each pixel by applying the radiative transfer equation to a sensor channel, which eventually leads to:

$$LST = \gamma \cdot \left[ \varepsilon^{-1} \left( \psi_1 \cdot L_{\text{sensor},\lambda} + \psi_2 \right) + \psi_3 \right] + \delta \qquad (15)$$

where $\gamma$ and $\delta$ are constants that depend on the top of atmosphere radiance ($L_{\text{sensor},\lambda}$) and the brightness temperature of the pixel, $\varepsilon$ is the emissivity of the pixel and $\psi_i$ are the atmospheric functions that are parameterized. More details are shown in Appendix A4 for completeness. The emissivity is considered equal to 0.98 on the whole scene, in line with our modelling chain. Water vapor and near-surface air temperature data come from ERA5 Reanalysis dataset (Muñoz Sabater, 2019). They are taken from the closest grid point. In order to cover a large range of solar zenith angles, a total of 20 cloudless thermal images from different winter dates were selected, from February 2015 to December 2019 (list in the appendix). The acquisition time of Landsat-8 observations (10h17 or 10h23 UTC depending on the scene) and in situ measurements (10h30 UTC, averaged over the previous 30 min) are considered to be equivalent.

## 3 Results

The spatially-resolved LST observations from Landsat-8 are first assessed in the study area, before the evaluation of the model simulations against the observations and the local measurements.





## 3.1 Surface temperature observations from Landsat-8

The surface temperature observations from Landsat-8 (LST) are compared to in situ FluxAlp measurements (Fig. 5a). The surface temperature from Landsat-8 is extracted from the pixel covering the location of FluxAlp from all 20 thermal images.

Both the single-channel (SC) method and the improved single-channel (iSC) method show better estimations at FluxAlp station than the official Collection 2 Surface Temperature product (-3.5 °C underestimation). As a result, we excluded this latter dataset from further analysis. The iSC method provides generally higher temperatures than the SC method, with a mean bias of -1.3 °C and -2.0 °C, respectively. Its total error is of 2.0 °C RMS, dominated by the bias (2.6 °C for the SC method). The improved single-channel method shows slightly more accurate results, and it is used in the following to evaluate the estimation of snow

surface temperature by the modelling chain.

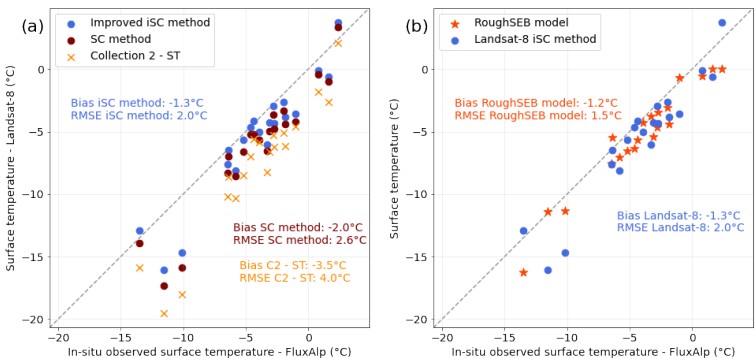

**Figure 5.** Landsat-8 retrieved surface temperature as a function of surface temperature measurements at FluxAlp station. In **(a)** using the iSC method (blue) and the SC method (brown), Collection 2 Surface Temperature product (orange); in **(b)** simulated $T_s$ by the model (red) and observed $T_s$ by Landsat-8 (iSC method – blue). The 1:1 dashed line represents perfect agreement with in situ measurements.

## 3.2 Snow surface temperature simulations

### 3.2.1 Validation at FluxAlp station

Figure 5b shows Landsat-8 observations and snow surface temperature simulations, compared to the in situ measurements at FluxAlp. The RoughSEB model is in general in agreement with the satellite observations. Considering all 20 scenes, they differ

by only 0.1 °C, the simulations being slightly warmer. The differences are however larger when considering the coldest cases, up to 5 °C. The error of the simulations is lower, of 1.5 °C RMS (2.0 °C for Landsat observations). The estimation of $T_s$ at this particular point is therefore accomplished for a variety of weather conditions and solar zenith angles.





### 3.2.2 Evaluation of the diurnal cycle

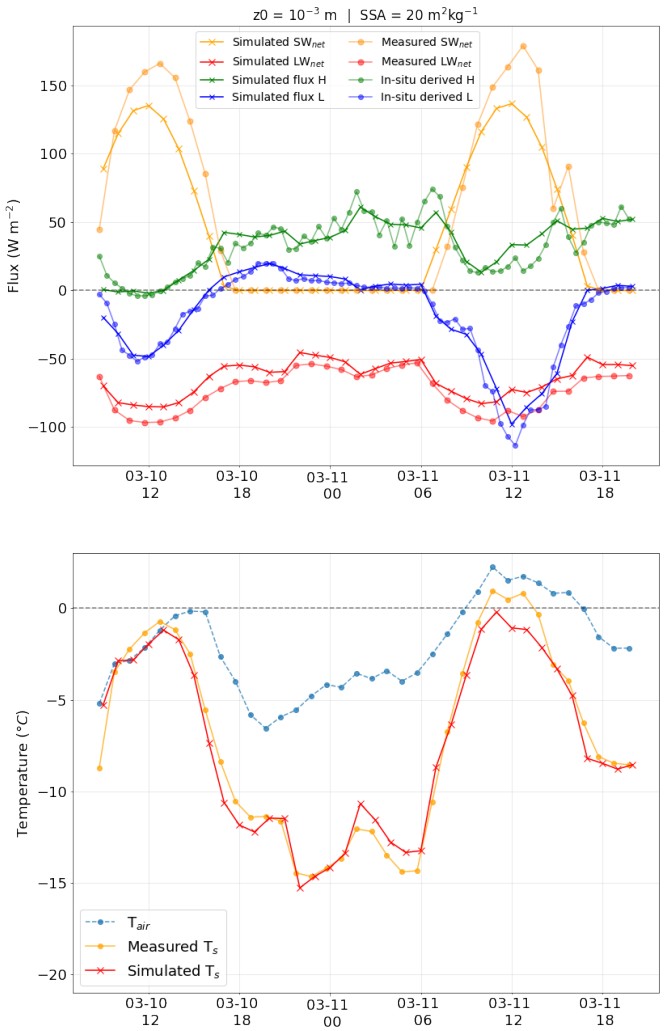

**Figure 6.** Simulation of the terms of the surface energy budget (top) and snow surface temperature (bottom) at the FluxAlp station for a $\approx$ 36 h long time series on March 2016. They are compared to in situ measurements (except $H$ and $L$ fluxes which are also estimated from the measurements). All times are in UTC.





The modelling chain to estimate $T_s$ is evaluated over a diurnal cycle at the FluxAlp station (Fig. 6). A period of $\approx 36\,\text{h}$
was selected after one of the Landsat-8 acquisition dates, starting at 9h UTC on 10 March 2016. This period featured stable
conditions and the sky was clear, except for a few minutes at the end of the period. Figure 6 (top) shows the temporal evolution
of the radiative fluxes ($\text{SW}_\text{net}$ and $\text{LW}_\text{net}$) and the turbulent fluxes (sensible heat flux $H$ and latent heat flux $L$). The simulations
are run hourly, with a constant $\text{SSA} = 20\,\text{m}^2\text{kg}^{-1}$ and aerodynamic roughness length $z_0 = 10^{-3}\,\text{m}$. They are compared to
the in situ measurements at the station, except in the case of the turbulent heat fluxes, which are as well simulated from the
measured wind speed, air temperature and humidity – from that the notation: in situ derived. As measurements at FluxAlp
site are averaged over the preceding 30 minutes, a 15 min time shift is applied on the graph for a fair comparison. The results
show that the net short-wave flux is underestimated except during the first hours of sunlight when it is overestimated. The
difference is small ($< 20\,\text{Wm}^{-2}$) around 10h-10h30 UTC, which by chance corresponds to the Landsat-8 acquisition time.
The underestimation leads to an overall bias during daytime of $\approx 15\,\text{Wm}^{-2}$, while the net long-wave radiation flux is in
general overestimated by $\approx 10\,\text{Wm}^{-2}$. The turbulent fluxes are well simulated compared to the values estimated at FluxAlp
site. It is worth recalling that both heat fluxes are calculated using the same equation and assumptions, which limits the
strength of this result. Figure 6 (bottom) shows the evolution of the simulated $T_s$ over the same period, compared to the in situ
measurements. Observed air temperature is also shown for completeness. There is a good agreement between the simulations
and the measurements, within -0.3 °C (RMSE: 0.9 °C). Surface temperature shows a remarkable diurnal cycle, where the
melting point is almost reached in the early afternoon, and the lowest temperatures are reached at night, in the absence of solar
radiation. The balance between the underestimation of the net SW flux (and therefore the energy absorbed in the snowpack)
and the overestimation of the net LW flux could explain the bias obtained in the snow surface temperature in the morning and
at the end of the afternoon. The slight variations in surface temperature (around 2 to 3 °C) during the night are mainly driven
by the balance between the long-wave radiation flux and the sensible heat flux, the other fluxes being negligible.

### 3.2.3 Evaluation of the spatial variations

To evaluate the spatial variations of $T_s$, the simulation corresponding to the Landsat-8 acquisition on 18 February 2018 is
analysed here. It is chosen because in situ SSA measurements were available (Tuzet et al., 2020), allowing a more effective
assessment of the surface albedo. The SSA value is $45\,\text{m}^2\text{kg}^{-1}$.

Figure 7 shows the spatial variations in snow surface temperature, observed by Landsat-8 (left) and simulated by the Rough-
SEB model (right). The variations are well represented by the model, with many similarities at all the scales across the scene.
The small-scale variations are better resolved by the model as its spatial resolution is significantly higher (10 m vs 30 m for
the satellite). This is in particular true in the northern area of the scene which covers a series of small valleys, showing a larger
temperature gradient in the simulation. The model is slightly colder in the coldest areas (e.g. shadows in the southern area
of the scene), as well as in the warmest areas around the FluxAlp station (green marker). As showed in Fig. 5, the surface
temperature does not differ significantly in the FluxAlp area.

Figure 8 shows the bias and standard deviation of a pixel-to-pixel comparison between all the 20 satellite observations and
the corresponding simulations. The latter are resampled to 30 m to match the spatial resolution of the Landsat-8 observations.

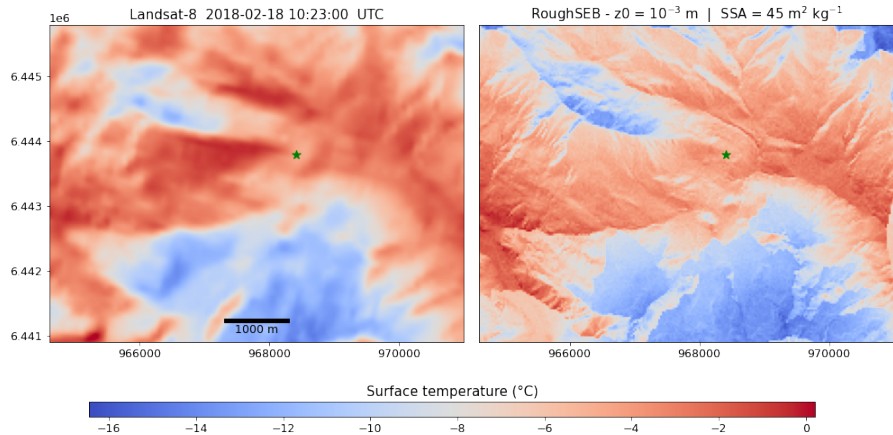

**Figure 7.** Surface temperature observed by Landsat-8 (left) and simulated by the RoughSEB model (right) in the Col du Lautaret area on the 18 February 2018. The location of the FluxAlp station is highlighted by the green marker. Projection is Lambert 93 (EPSG: 2154) and the coordinates are in meters.

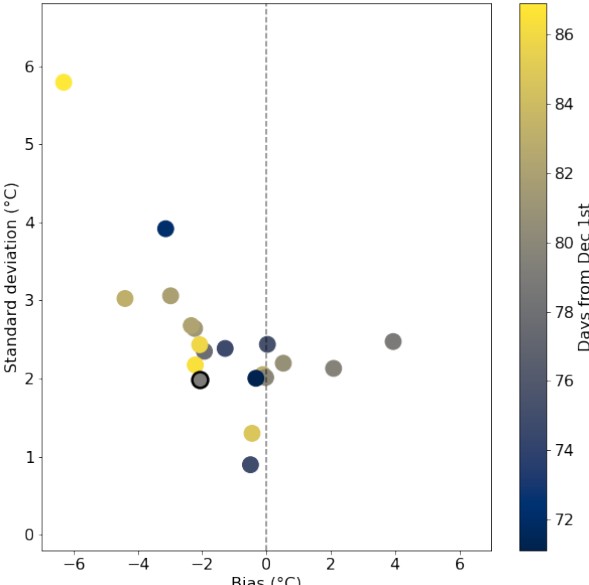

**Figure 8.** Mean bias and standard deviation between the simulations and the satellite observations for each date, computed considering the whole area. The simulation on 18 February 2018 presented in Fig. 7 is highlighted by the rounded marker.

The simulations are colder in general, with a negative bias comprised principally between -4 °C and 0 °C. Some simulations are however warmer than the satellite observations. The standard deviation of the difference varies mostly between 2 and 4 °C.





The simulation that shows the highest differences corresponds to an acquisition from late March 2019. Such differences could
be explained with an early onset of snowmelt (snow patches in the lowest areas), a particular situation that is not yet correctly
resolved by the model. Nevertheless, considering all acquisitions, these differences don't seem to be clearly related to the time
of the year (i.e. early or late winter), so strong conclusions cannot be drawn.

This results show the performance of the RoughSEB model to simulate the snow surface temperature and the surface energy
budget in complex terrain within a reasonable accuracy. Their temporal and spatial variations are also well represented in the
study area. To understand these variations, the role of the topographic effects that govern them is addressed in the following
section.

### 3.2.4  Relative importance of the topographic effects

In order to quantify the relative importance of the topographic effects, we have run a series of simulations where every effect
considered in this work is disabled at a time. Figure 9 shows the impact on the surface temperature distribution across the area
for the 18 February 2018. The scatter plots show the modelled $T_s$ for every pixel in the area as a function of the $T_s$ observed
by Landsat-8. The histogram plots show the distribution of all the pixels and in addition, the reference simulation (black line)
– including all the effects – and Landsat-8 observations (red).

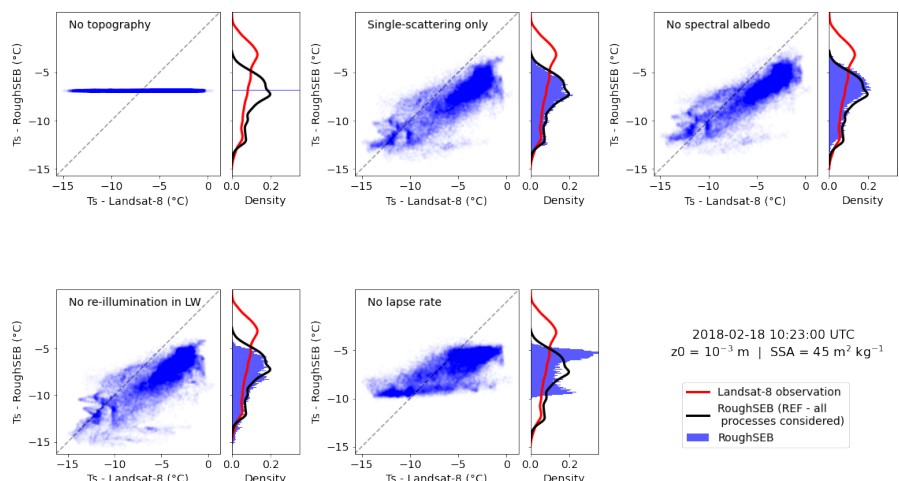

**Figure 9.** The simulated $T_s$ on 18 February 2018 as a function of the observed $T_s$ by the satellite. Every single panel correspond to a
topographic effect disabled, with respect to the reference simulation (REF) where all processes are considered. The marginal histograms
show the distribution of surface temperature for each simulations as well as the observed $T_s$ (red) and the reference simulation (black).

To extend these results to all the dates, Fig. 10 displays summary statistics of the difference between the simulations with a
disabled topographic effect and the reference simulation. The left panel presents the mean difference between the simulation
without the effect and the reference simulations (for each effect) while the standard deviation of the differences (right panel)





shows how both simulations agree in terms of spatial variations. A value close to zero means that the effect is negligible in terms of average and variations respectively.

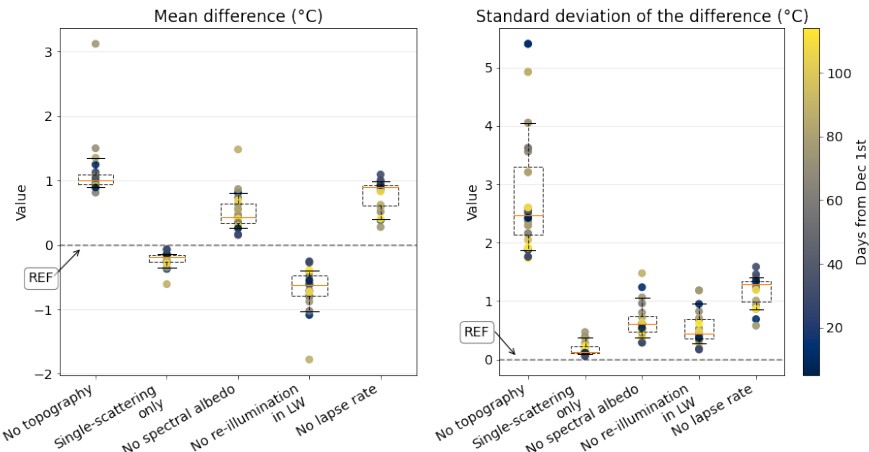

**Figure 10.** Overall representation of the mean difference and standard deviation between the reference simulation (REF), where all processes are considered, and the additional simulations where one topographic effect is disabled at a time. The whiskers of the boxplots represent the 10th and 90th percentiles of the distribution.

When no topography is taken into account (i.e., a perfectly flat surface instead of the actual topography), the snow surface temperature is uniform (-6.9 °C) across the scene. This value overestimates the mean temperature of the reference simulation (-7.9 °C) showing that not only the variations of temperature are not represented but even on average a simulation on a flat terrain is not equivalent to the mean temperature of the area with topography. This basic simulation highlights the considerable effect of the topography on the surface temperature and the importance to take it into account even for large scale simulations. Considering all the 20 dates, neglecting the topography results in an overall overestimation of 1.0 °C (median value) and the standard variations with respect to the reference simulation are high with a median value of 2.5 °C and a maximum of 5 °C.

The simulation with single-scattering only in the SW, (only the first bounce of the photons is considered by RSRT, multiple scattering is neglected) shows small differences with respect to the reference simulation. It is only slightly colder, in particular the coldest pixels. This is mainly due to neglecting the re-illumination caused by multiple bouncing in the shadowed (and cold) areas. Nevertheless, the impact is not significant, the mean difference being of barely different from zero (median of -0.2 °C, standard variations of 0.1 °C). The smallest impact is observed in December which could be explained by a dependence on the solar zenith angle, and on how the topography modulates the received solar irradiance. Overall, this result means that, at least in our study area, the absorption enhancement caused by multiple bouncing is almost negligible.

When neglecting the spectral dependence of snow albedo and thus considering only the broadband albedo ($\approx 0.87$ on 18 February 2018), the simulated surface temperature is slightly warmer than the reference with overall differences of 0.4 °C (median of the means) and of 0.6 °C (median of standard deviations) and a maximum beyond 1 °C. This effect involving the





coupling between spectral dependence of snow albedo and topographic effect is important to take into account, as it plays a role that needs to be considered.

The effect of the thermal emission by surrounding terrain in the LW is obtained by estimating the surface temperature as if the terrain was flat (first step as detailed in Sect. 2.1.2). The peak of the distribution is less marked and widened in the range

(-5 °C – -8 °C), and the impact is a systematic underestimation of $T_s$. This is true for the warmest areas of the scene (higher than - 5 °C) where the underestimation is of -0.7 °C, and in particularly true for the coldest areas (lower than -12 °C), with an underestimation of -1.2 °C. In the latter, direct radiation is absent and the radiative budget is dominated by diffuse and weak illumination. The thermal emission by surrounding faces warms these cold areas as a function of their sky-view factor. The mean difference goes down to -0.6 °C (standard variations of 0.4 °C) when considering all the scenes.

With respect to the altitudinal variations of air temperature (lapse rate effect), the distribution shape squeezes and even becomes bimodal, with two marked peaks at -5 °C and -10 °C. The difference to the reference simulation is significant when considering all the dates, with a median value of 0.9 °C and a median standard deviation of 1.3 °C. Introducing the lapse rate effect tends to warm the air on the lower parts of the scene (usually the warmer) and the opposite on the higher and colder facets. Since the FluxAlp measurement station (where the reference air temperature is taken) is in the lower range of altitudes

of the study area the overall impact neglecting the lapse rate effect is an overestimation (0.9 °C). This result is specific to our setting and using a different reference air temperature would change this result. In principle it is possible to choose the reference at the mean altitude of the area which leads to a null bias. Nevertheless, the standard variations are also significant and this is not specific to our setting. It effectively shows that neglecting the altitudinal variations of air temperature results in an overall significant error of 1.3 °C (median) in surface temperature over the study area.

Theses results gives the relative importance of the topographic effects investigated here. Neglecting the topography (i.e. a flat surface) is, as expected, the most important source of error when simulating snow surface temperature. Neglecting the altitudinal variations of air temperature (lapse rate effect) is the second effect in terms of importance. It is followed by the thermal emission by surrounding terrain in the LW and the spectral dependence of snow albedo, the latter being slightly less important. Finally, the absorption enhancement caused by multiple bouncing is almost negligible.

## 4 Discussion

### 4.1 Retrieval of surface temperature from satellite observations

The assessment of our modelling chain was performed against in situ and satellite observations, the latter being crucial for the spatial variations. However, these depend on the choice of the method for the atmospheric correction. Here we implemented two single-channel methods for Landsat-8 thermal imagery: the SC method (Jiménez-Muñoz and Sobrino, 2003), and the

iSC method (Cristóbal et al., 2018) and we compared them to the recent Surface Temperature product. The iSC method is the most accurate (within ≈ 1 °C) at the FluxAlp measurement station. The Collection 2 Surface Temperature product is the less accurate (within 3.5 °C). However, this does not preclude of the quality in the whole area. In particular, we assumed that the whole scene is covered by snow, with an emissivity equal to 0.98. However, in alpine areas, vertical mountain ridges or





patches without snow on sun-facing slopes are frequent and may results in more variable emissivity over the scene. A possible
future improvement would be to consider an emissivity mask, where each pixel would have a particular value depending on
the presence of snow, rocks, grass, etc. This is normally achieved by means of NDVI-based classifications (Li et al., 2013).
The Surface Temperature product (which is already generated using this method) presents worse results at our validation point,
but overall, the differences across the study area between the iSC method and this official product are of 0.3 °C (median of
standard deviations) considering the whole dataset, which is considerably lower than the difference at FluxAlp. The satellite
thermal observations from Landsat-8 are therefore correct enough to assess the model performance, with an accuracy of the
order of 1 °C and a precision lower than 0.5 °C.

## 4.2 Snow surface temperature estimation

One question that arises is about the performance of the model to estimate the $T_s$ and its temporal and spatial variations in
complex terrain. The simulations show an overall agreement with measurements and satellite observations (Sect. 3.2). The
simulated fluxes to be considered in the surface energy budget and the temporal evolution of $T_s$ are well represented for a
daily cycle (Fig. 6). The net SW flux is slightly underestimated during most of the day (except in the early morning when it is
slightly overestimated), but the impact on surface temperature is balanced by the other terms of the energy budget, in particular
due to the slight overestimation of the net LW flux. The turbulent heat fluxes are essential for an accurate calculation of the
surface energy balance, in particular during the night, when they balance the LW radiation flux. Their treatment in the model is
very simple, the atmosphere is assumed neutral, the aerodynamic roughness length is uniform, the wind is uniform, etc. Some
of these simplifications are required to achieve good computational performances and may be challenging to take into account
(wind field) while others (e.g. atmospheric stability) could be easily implemented (Essery and Etchevers, 2004).

The ground heat flux was also neglected meaning that the surface temperature reacts immediately to changes of the down-
welling radiation. In reality, the snowpack has some thermal inertia which delays and moderates the diurnal amplitude of $T_s$,
resulting in an overestimation in the morning and underestimation in the late afternoon, when the cooling of the surface is
more pronounced in the simulations as the solar zenith angle increases. Nevertheless, this delay is barely visible in our case
(Fig. 6). Snow is indeed a highly insulating medium and has a small thermal inertia. With a thermal conductivity of around
$0.2\,\mathrm{Wm^{-1}K^{-1}}$ (Sturm et al., 1997), the daily wave penetrates by no more than $20\,\mathrm{cm}$ in the snowpack. It means that with a
diurnal cycle half-amplitude of $\approx 7\,°\mathrm{C}$ in surface temperature, the maximum temperature gradient in the upper snowpack is of
the order of $35\,\mathrm{Km^{-1}}$. According to the Fourier law, this implies a maximum ground heat flux $G \approx 7\,\mathrm{Wm^{-2}}$ which is an order
of magnitude lower than the radiative and turbulent heat fluxes estimated in our case (Fig.6). This simple calculation confirms
that neglecting $G$ is acceptable in a first approximation when snow is relatively fresh and is not melting. In spring, with a
denser and less insulating snowpack and with melt, this approximation needs to be reconsidered. However, taking into account
the thermal diffusion in the snowpack over millions of facets represent a very significant computational cost and requires
approximations.

The choice of input parameters such as SSA and aerodynamic roughness length ($z_0$) is critical for the simulations. Local
measurements are not always available and may not be representative of the whole area. SSA plays a crucial role on the albedo





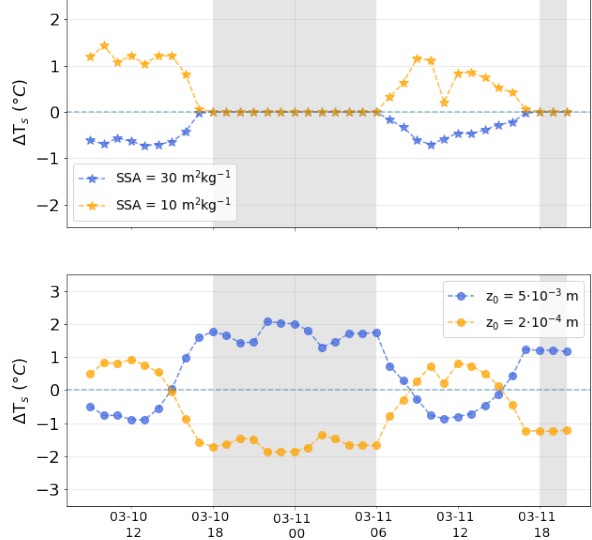

**Figure 11.** Changes in simulated snow surface temperature ($\Delta T_s$) for a diurnal cycle at the FluxAlp station when varying the specific surface area (SSA – top) and the aerodynamic roughness length ($z_0$ – bottom). The shaded areas corresponds to night time (i.e. SZA below 0° or over 90°).

and so on the short-wave absorption by the snowpack. The sensitivity of snow surface temperature to SSA is shown in Fig. 11 (top), for the diurnal cycle presented in Fig. 6. SSA varies in $\pm\ 10\,\mathrm{m^2kg^{-1}}$ with respect to the reference simulation for

the whole time series ($20\,\mathrm{m^2kg^{-1}}$). The impact of varying SSA on $T_s$ is up to $1\,°\mathrm{C}$, and is larger when considering a low SSA than a large SSA. This is mainly due to the fact that the relationship SSA-albedo is not linear, as shown by Domine et al. (2006). In general spatio-temporal variations of SSA are expected in our area because the initial SSA value depends on snowfall temperature mainly and its subsequent evolution is a general decrease depending on the thermal evolution of the snowpack (Domine et al., 2008). SSA is expected to be higher at higher altitudes and in the shadows where the conditions are

colder, and so does the short-wave absorption which would tend to increase the cooling in these areas compared to lower SSA areas. This could also explain the differences between the simulated and measured net short-wave flux, as here the value was kept constant for the whole time series.

      The aerodynamic roughness length controls the sensible and latent heat fluxes, so it shall also be carefully tuned. In this study, a value of $10^{-3}$ m is assumed, which is standard according to previous works (e.g. Brock et al. (2006)). The sensitivity

of $T_s$ to aerodynamic roughness length is shown in Fig. 11 (bottom). $z_0$ varies by a factor of 5 with respect to the reference simulation. The choice of $z_0$ has a more significant impact on the simulated surface temperature, in particular during the night (up to $2\,°\mathrm{C}$), when the SW radiation is absent. Martin and Lejeune (1998) found similar variations using the snow model Crocus, with a slightly different approach that considered changes of atmosphere stability, as well as Kuipers Munneke et al. (2009) with even another approach for the turbulent heat fluxes. According to Brock et al. (2006), snow $z_0$ can vary up to three





orders of magnitude depending on the time of the season (i.e. early or mid ablation season) and also on snow type (i.e. fresh snow or melting snow). To estimate its expected spatial variations in a mountainous area of several $km^2$, a parametrization of SSA evolution would be possible (Domine et al., 2007) or by using SSA retrieved by satellite (e.g. Kokhanovsky et al. (2019)).

    The spatial variations of surface temperature are clearly dependent on the topography and are correctly simulated (Fig. 7). They seem to depend in particular on slope orientation, as shown in Fig. 12. For larger slopes (> 30°), the lack of direct
radiation governs the surface temperature in some areas, such as the northern (N-W to N-E) slopes covered in the shadow at the southern part of the scene. The south-facing slopes are more numerous and feature an extended range of surface temperature. The temperature difference is large between opposed slopes (on the order of 5 to 10 °C in a few hundred meters). Overall, the mean $T_s$ of the northern facets is -10.5 °C (± 1.7 °C). They are considerably colder than the southern (S-E to S-W) ones (-6.7 °C (± 1.5 °C)). These differences are consistent with previous studies (e.g. Fierz et al. (2003)), and highlight the necessity
of accounting for spatial variations of surface temperature in mountainous areas, where larger slopes prevail over gentle terrain.

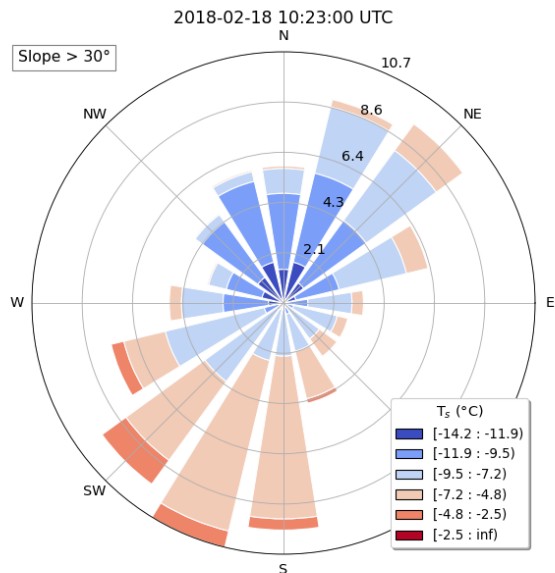

**Figure 12.** Distribution of simulated surface temperature as a function of the aspect for slopes larger than 30°. The radius of each of the 16 sectors corresponds to the normed (displayed in percent) quantity of facets.

### 4.3   Role of the topographic effects

The results presented in this study show that the topography controls the energy budget and the surface temperature to a large extent. However modelling all the processes involved in complex terrain may lead to prohibitive computational cost for most applications. This has an adverse outcome: some of them are usually neglected or approximated. Our modelling chain takes
into account a relatively comprehensive set of processes, especially on the radiative aspects. The motivation of this study is



therefore to quantify their relative importance, in order to provide some insight on which one of them can be neglected as a function of the targeted accuracy.

The role of the topography is quantified in Sect. 3.2.4, where we compared the reference simulation (where all effects are considered) to simulations where we disabled one effect at a time. Overall, we found that the absence of topography (i.e. terrain
altitude, slope and aspect), and therefore the presence of altitudinal variations and self and cast shadows is the most important effect. Errors of several °C are to be expected if considering a flat surface. The same conclusion is drawn by Yan et al. (2016) which found significant errors in the estimation of net LW radiation flux by assuming a flat surface at a larger scale over the Tibetan Plateau.

Yan et al. (2016) also accounted for the sky-view factor and the contribution of thermal radiation coming from surrounding
terrain. Our results show a cooling effect of mean difference of almost 1 °C (median of -0.6 °C) when disabling these topographic effects. This agrees with the results presented by Arnold et al. (2006), showing a similar ratio of importance between the role of the shadows and the LW contribution when considering glacier melt. To take into account this effect, the downwelling LW radiation flux in each facet is updated with the thermal emission of the surrounding facets, which is derived with an average $T_s$ for the whole scene. Such an approximation saves a lot of computation time but, as a result, the differences in
thermal emission due to spatial variations of surface temperature are masked. The warmest facets will emit more LW radiation than the coldest ones, leading to differences in the modelled $T_s$.

Our results show a significant impact on the simulated $T_s$ by disabling the altitudinal variations of air temperature, with a warming effect up to 1 °C. The spatial and temporal distribution of air temperature, as well as wind dynamics in mountainous areas, have been widely investigated (Jiménez and Dudhia, 2012; Rotach et al., 2015), and different parameterization and
estimation methods exist to overcome its complexity (e.g. Wood et al. (2001)). The approach implemented here is simple, with several approximations and important assumptions as a constant aerodynamic roughness length and wind speed across the study area. Future improvements of our approach could benefit from downscaling methods to better represent wind distribution over complex terrain, as in Helbig et al. (2017).

Arnold et al. (2006) also pointed out the role of the anisotropic reflectance of snow and ice at high solar zenith angles. This
is found to be essential to correctly estimate the radiative budget. The RSRT model can account for this effect (Larue et al., 2020) but we did not consider it here. The reason is that we preferred to account for the spectral dependence of snow albedo, which is an improvement with respect to most of the models that only account for broadband albedo. The impact in $T_s$ is however limited, but errors up to $\approx 1$ °C are possible. As running RSRT at many wavelengths is very expensive, we developed the effective method present in Eqs. 5 and 6. Unfortunately, this method relies on the Lambertian approximation which makes
it incompatible with taking into account the reflectance anisotropy.

Another topographic effect investigated here is the influence of the multiple bounces of photons between the faces in the short-wave domain, also called re-illumination or absorption enhancement. It is simulated with the RSRT model (Larue et al., 2020) and represents the most expensive step of the modelling chain in terms of computational cost, as each photon is tracked until it escapes the scene, after one or several bounces. Our results suggest that the re-illumination can be neglected, with a
limited impact on the surface temperature estimations (< 0.3°C). This implies that RSRT is only needed to track the shadows,





slope and compute the sky view factor. In principle these effects can be taken into account by other methods (e.g. Dozier et al. (1981)). Nevertheless, other authors have drawn different conclusions regarding the role of re-illumination in the same study area. Lamare et al. (2020) found a significant contribution of multiple reflections on the simulated TOA radiance in rugged terrain. An hypothesis to explain this discrepancy is the dependence on the sun's position and the configuration of the terrain.

Earlier simulations in the winter season (closer to December 1st) show a lower impact than later ones. However, this point requires further investigation to derive stronger conclusions, in particular as atmospheric scattering and absorption effects due to the air are neglected in the present work.

## 5 Conclusions

We have investigated the relative importance of several topographic effects in a mountainous terrain in the French Alps.

For this, we have first developed a chain of models to predict surface temperature, by combining existing radiative transfer models (RSRT, SBDART) and a new surface energy budget model (RoughSEB). This chain has been evaluated against in situ measurements and remote sensing thermal observations to account for the spatial variations. The latter have been corrected for atmospheric effects with a single-channel algorithm.

A $\approx$ 36 h long time series was simulated and an overall agreement is achieved with the in situ measurements, within 0.3 °C.

Besides, the bias of the simulations at FluxAlp station for the 20 scenes corresponding to the Landsat-8 acquisitions is only -1.2 °C (total error of 1.5 °C RMS), which highlights the potential of the chain to simulate surface temperature within a reasonable accuracy and precision. The spatial variations across the $50 \, \text{km}^2$ scene are also well represented, with a standard deviation of the differences to the satellite observations comprised between 2 and 3 °C, which is small compared to the actual surface temperature variations of 5-10 °C between the slopes in the study area.

A few topographic effects that are responsible for such spatio-temporal variations have been investigated and their relative importance is: 1) The modulation of solar irradiance by the terrain slope and aspect (i.e. the presence of topography), 2) the altitudinal changes in air temperature (lapse rate effect), 3) the contribution of thermal radiation coming from surrounding terrain, 4) the spectral dependence of snow albedo, and 5) the absorption enhancement caused by multiple bouncing of photons in the SW domain. Their importance has been quantified and the warming (or cooling) effects are up to 1 °C, except for the

first one (absence of topography) that lead to errors of several degrees Celsius in surface temperature.

The modelling chain shows some limitations justified by the assumptions made on some of the parameters controlling the radiative budget, such as snow SSA or the aerodynamic roughness length $z_0$. Nevertheless, it shows good performance to estimate snow surface temperature and its spatial variations, and has several applications. It allows a better understanding of the processes that govern the surface energy budget in snow-covered, mountainous areas. A first extension of the model is

to investigate the snowmelt. Another application is the preparation, and in the future the calibration/validation of the thermal infrared TRISHNA satellite mission (Lagouarde et al., 2019) which will provide from 2025 high resolution images (50 m) of the Earth surface temperature in mountainous areas.





**Appendix A: Additional equations and mathematical developments**

**A1 Snow spectral albedo in the Asymptotic Radiative Transfer (ART) theory**

The direct and diffuse components of the snow spectral albedo are computed using the ART theory (Kokhanovsky and Zege, 2004). Provided several assumptions about the snowpack (semi-infinite, vertical and horizontal homogeneous layers) and the surface (flat and smooth – facets are small enough to be considered as it), they are expressed as follows:

$$\alpha_{\mathrm{dir}}(\lambda, \theta_{\mathrm{s}}) = \exp\left(-\frac{12(1+2\cos\theta_{\mathrm{s}})}{7}\sqrt{\frac{2B\gamma(\lambda)}{3\rho_{\mathrm{ice}}\mathrm{SSA}(1-g)}}\right) \tag{A1}$$

$$\alpha_{\mathrm{diff}}(\lambda) = \exp\left(-4\frac{\sqrt{2B\gamma(\lambda)}}{3\rho_{\mathrm{ice}}\mathrm{SSA}(1-g)}\right) \tag{A2}$$

where $\theta_{\mathrm{s}}$ is the solar zenith angle, SSA is the snow specific surface area, $\rho_{\mathrm{ice}} = 917\,\mathrm{kg\,m^{-3}}$ is the bulk density of ice at $0\,^{\circ}\mathrm{C}$, $\gamma(\lambda)$ is the ice absorption coefficient (from Picard et al. (2016)) and $B$ and $g$ are the shape coefficients of snow, taken from Libois et al. (2014).

**A2 Surface exchange coefficient**

The surface exchange coefficient involved in the calculation of the turbulent heat fluxes is given by:

$$C_{\mathrm{H}} = 0.16\left[\ln(\frac{z_{\mathrm{t}}}{z_0})\ln(\frac{z_{\mathrm{w}}}{z_0})\right]^{-1} \tag{A3}$$

where $z_t$ and $z_w$ are the height at which air temperature and wind speed are measured, and $z_0$ is the aerodynamic roughness length. Their values are provided in Table B1.

**A3 General solution to the quartic equation for $T_{\mathrm{s}}$**

The simplified surface energy budget equation eventually become a quartic equation for $T_{\mathrm{s}}$ of the form:

$$a\,T_{\mathrm{s}}^4 + d\,T_{\mathrm{s}} + e = 0 \tag{A4}$$

where $a$, $d$, $e$ are a constant (for each facet). The general solution can be calculated using:

$$T_{\mathrm{s}} = -S + \frac{1}{2}\sqrt{-4S^2 + \frac{q}{S}} \tag{A5}$$





where

$$q = \frac{d}{a} \tag{A6}$$

$$\Delta_0 = 12\, a\, e \tag{A7}$$

$$\Delta_1 = 27\, a\, d^2 \tag{A8}$$

$$Q = \sqrt[3]{\frac{\Delta_1 + \sqrt{\Delta_1^2 - 4\Delta_0^3}}{2}} \tag{A9}$$

$$S = \frac{1}{2}\sqrt{\frac{1}{3a}\left(Q + \frac{\Delta_0}{Q}\right)} \tag{A10}$$

$$\tag{A11}$$

## 570 A4    Land Surface Temperature retrieval from Landsat 8

LST is calculated for each pixel by applying the radiative transfer equation to a sensor channel, which eventually leads to:

$$LST = \gamma \cdot \left[ \varepsilon^{-1}\left(\psi_1 \cdot L_{\text{sensor},\lambda} + \psi_2\right) + \psi_3 \right] + \delta \tag{A12}$$

where:

$$\gamma = \left\{ \frac{c_2 \cdot L_{\text{sensor},\lambda}}{T_{\text{sensor}}^2} \left[ \frac{\lambda^4}{c_1} L_{\text{sensor},\lambda} + \lambda^{-1} \right] \right\}^{-1} \tag{A13}$$

$$\delta = -\gamma \cdot L_{\text{sensor},\lambda} + T_{\text{sensor}} \tag{A14}$$

$$T_{\text{sensor}} = \frac{K_2}{ln\left(\frac{K_1}{L_\lambda} + 1\right)} \tag{A15}$$

where $\varepsilon$ is the emissivity of the pixel, $\psi_i$ are the atmospheric functions that are parameterized, $\lambda$ is the effective wavelength (10.904 μm for Band 10), $L_{\text{sensor},\lambda}$ is the top of atmosphere radiance calculated from pixel Digital Numbers (DN) using rescaling factors (USGS, 2021), and $T_{\text{sensor}}$ is the brightness temperature in Kelvin. Values of the symbols can be found in Table B1.

The atmospheric functions are statistically fitted from the GAPRI database (Mattar et al., 2015) containing 4714 atmospheric profiles from tropical to arctic atmospheric conditions. The following fit is applied here:

$$\psi_i = i\, w^2 + h\, T_a^2 + g\, w + f\, T_a + e\, T_a^2\, w + d\, T_a\, w + c\, T_a\, w^2 + b\, T_a^2\, w^2 + a \tag{A16}$$

All the coefficient values (from $i$ to $a$) are in Cristóbal et al. (2018).



## Appendix B: Table of symbols

**Table B1.** Definitions and values of the symbols and magnitudes that appear in this manuscript.

| Symbol | Description | Value |
|:---:|:---:|:---:|
| $\sigma$ | Stefan-Boltzmann constant | $5.67 \cdot 10^{-8} \, \mathrm{Wm^{-2}K^{-4}}$ |
| $P_s$ | Air pressure | Altitude dependent [Pa] |
| $\rho_{air}$ | Air density | $P_s \cdot (287 \, T_{air})^{-1} \, [\mathrm{kgm^{-3}}]$ |
| $c_{p,air}$ | Heat capacity of air | $1005 \, \mathrm{Jkg^{-1}K^{-1}}$ |
| $L_s$ | Sublimation heat | $2.838 \cdot 10^6 \, \mathrm{Jkg^{-1}}$ |
| $z_t$ | Temperature measurement height | $3.53 \, \mathrm{m}$ - snowdepth [m] |
| $z_w$ | Wind speed measurement height | $5.18 \, \mathrm{m}$ - snowdepth [m] |
| $z_0$ | Roughness length | $10^{-3} \, \mathrm{m}$ |
| $c_1$ | Planck radiation constant | $1.19104 \cdot 10^8 \, \mathrm{W\mu m^4 m^{-2} sr^{-1}}$ |
| $c_2$ | Planck radiation constant | $1.43877 \cdot 10^4 \, \mathrm{\mu m \, K}$ |
| $K_1$ | Landsat calibration constant | $774.89 \, \mathrm{Wm^{-2}sr^{-1}\mu m^{-1}}$ |
| $K_2$ | Landsat calibration constant | $1321.08 \, \mathrm{K}$ |





## Appendix C: List of selected Landsat-8 scenes

**Table C1.** List of selected scenes

| Date | Path / Row | Product name |
|---|---|---|
| 10 February 2015 | 196 / 029 | LC08_L1TP_196029_20150210_20170413_01_T1 |
| 19 February 2015 | 195 / 029 | LC08_L1TP_195029_20150219_20170412_01_T1 |
| 26 February 2015 | 196 / 029 | LC08_L1TP_196029_20150226_20170412_01_T1 |
| 21 January 2016 | 195 / 029 | LC08_L1TP_195029_20160121_20170405_01_T1 |
| 9 March 2016 | 195 / 029 | LC08_L1TP_195029_20160309_20170328_01_T1 |
| 13 December 2016 | 196 / 029 | LC08_L1TP_196029_20161213_20170316_01_T1 |
| 1 January 2018 | 196 / 029 | LC08_L1TP_196029_20180101_20180104_01_T1 |
| 2 February 2018 | 196 / 029 | LC08_L1TP_196029_20180202_20180220_01_T1 |
| 18 February 2018 | 196 / 029 | LC08_L1TP_196029_20180218_20180307_01_T1 |
| 27 February 2018 | 195 / 029 | LC08_L1TP_195029_20180227_20180308_01_T1 |
| 22 March 2018 | 196 / 029 | LC08_L1TP_196029_20180322_20180403_01_T1 |
| 4 January 2019 | 196 / 029 | LC08_L1TP_196029_20190104_20190130_01_T1 |
| 29 January 2019 | 195 / 029 | LC08_L1TP_195029_20190129_20190206_01_T1 |
| 5 February 2019 | 196 / 029 | LC08_L1TP_196029_20190205_20190221_01_T1 |
| 14 February 2019 | 195 / 029 | LC08_L1TP_195029_20190214_20190222_01_T1 |
| 21 February 2019 | 196 / 029 | LC08_L1TP_196029_20190221_20190308_01_T1 |
| 18 March 2019 | 195 / 029 | LC08_L1TP_195029_20190318_20190325_01_T1 |
| 25 March 2019 | 196 / 029 | LC08_L1TP_196029_20190325_20190403_01_T1 |
| 6 December 2019 | 196 / 029 | LC08_L1TP_196029_20191206_20191217_01_T1 |
| 31 December 2019 | 195 / 029 | LC08_L1TP_195029_20191231_20200111_01_T1 |



*Author contributions.* AR and GP designed the study. GP and FL wrote RSRT, and IO, GP and FL wrote an initial version of RoughSEB. All co-authors contributed to the development of RoughSEB and to the analysis and interpretation of the results. AR prepared the manuscript with contributions of all co-authors.

*Competing interests.* The authors declare that they have no conflict of interest.

*Acknowledgements.* This research has been supported by the Centre National d'Études Spatiales (MIOSOTIS) and the Agence Nationale de la Recherche (ANR-19-CE01-0009 MiMESis-3D). It was also supported by Lautaret Garden – UMS 3370 (Univ. Grenoble Alpes, CNRS, SAJF, 38000 Grenoble, France), a member of AnaEE France (ANR-11-INBS- 0001 AnaEE-Services, Investissements d'Avenir frame) and of the eLTER European network (Univ. Grenoble Alpes, CNRS, LTSER Zone Atelier Alpes, 38000 Grenoble, France). The authors would like to acknowledge staff at Institut des Géosciences de l'Environnement for providing FluxAlp measurements and EBONI team (ANR-16-
CE01-0006) for providing SSA measurements.



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
