# Peer review of "Modelling surface temperature and radiation budget of snow-covered complex terrain"

_The Cryosphere, 2021_

## Referee Comment (RC1)

**Review report for the manuscript-**
**Robledano et al., "Modelling surface temperature and radiation budget of snow-covered complex terrain"**

The authors claim to estimate the LST and the the energy budget of snow-covered complex terrains, in order to evaluate the significance of the different processes in influencing the spatial variations of the LST. The strategic analysis is interesting and significant for the scientific community. However, some issues remain to be discussed and some revisions are required before the manuscript could be accepted for publication. My specific comments are as follows.

1. The authors claim to estimate the energy budget of the snow-covered complex terrains. However, this is not discussed in the results.
2. Why the double channel method (spit window) is not examined by the authors?
3. The discussion in the results is mostly qualitative. The analysis lacks statistical depth. The discussions include at places standard deviation, mean difference etc., but not popular metrics such as the correlation coefficient and the RMSE. For example in Figure 8, 9 and 10.
4. The authors mentioned and illustrated the effects of the topography on the estimated LST. However, they did not consider any literature on orographic corrections. For example as follows which is replicable for LST in a similar manner.

   Bento, V.A.; DaCamara, C.C.; Trigo, I.F.; Martins, J.P.A.; Duguay-Tetzlaff, A. Improving Land Surface Temperature Retrievals over Mountainous Regions. Remote Sens. 2017, 9, 38. https://doi.org/10.3390/rs9010038

   Varade, D., & Dikshit, O. (2019). Improved assessment of atmospheric water vapor content in the Himalayan regions around the Kullu Valley in India using Landsat-8 data. Water Resources Research, 55, 462– 475. https://doi.org/10.1029/2018WR023806

5. The authors mentioned the limitations of the NDVI thresholds method for the estimation of emissivity. The authors may explore the following alternative,

   Divyesh Varade & Onkar Dikshit (2020) Assessment of winter season land surface temperature in the Himalayan regions around the Kullu area in India using landsat-8 data, Geocarto International, 35:6, 641-662, DOI: 10.1080/10106049.2018.1520928

   Further, the authors missed the influence of the vegetation or the forest cover in their analysis, which is significant on the LST and the atmospheric water vapor content.

6. Since, the comparison is made against the Landsat-8 derived LST, it is imperative that the used reference product is at the most best quality. I would recommend the authors to calibrate this product from a series of ground station data if available.

7. Comments regarding the write-up.
    a) The language of the manuscript is extremely poor. It is difficult to understand because of the poor language used. The following checks are required by the authors
        i. Missing punctuations. Example- Line 1,5 in abstract.
        ii. Grammatical mistakes, usage of incorrect articles.
        iii. Usage of appropriate words. For example, Line 28, "Terrain tilt", I believe should be "Terrain orientation" or "Terrain slope" . The sentence is very difficult to understand and there are several such sentences in the manuscript.
        Another example, Line 214 it should be "quadratic". And so on.
    b) Figure 3, instead of showing the chart in the left image, the authors can show the slopes and their directions using directional gradient filters applied on the DEM.
    c) Abbreviations/symbols needs to be defined in Figures, For example in Figure 1 and 2. In some cases, the definitions of these come after several paragraphs or in other sections.
    d) The discussion of the some of the Figures and corresponding results is not sufficient. For example, in Figure 8, bulk of the points are between $\sigma$ of ~1-3 $^o$C, Hoverever, some outliers are also observed. These are not discussed in the manuscript. Any particular reasons for this.
    e) Figure 7, it would be interesting to see how a downscaled Landsat-8 LST would fare against the results from the proposed methodology.

---

## Referee Comment (RC2)

Review of Robledano et al. "Modelling surface temperature and radiation budget of snow-covered complex terrain"

The topic of the manuscript is very interesting and challenging. The authors claim that their developed modelling procedure, which involves several steps and the use of different schemes, enables the calculation of the surface temperature and surface energy budget over snow-covered mountain areas at high spatial resolution (10m). Although some of the results seem encouraging, the method is not clearly described, and there are macroscopic shortcomings in the modelling, the biggest ones being that the applied downward longwave radiation is not corrected for variations in altitude, and that the equation to extract the surface temperature from the surface energy balance equation is totally obscure and seems rather arbitrary (looks like that the Ts dependency on air specific humidity and shortwave flux are ignored?). Also, the solar infrared flux for wavelengths longer than 2000 nm is neglected, without explaining neither the reason for the neglection nor the implications of this neglection in the results (actually, this is the case also for other applied approximations). Finally, the model seems applicable only for clear-sky conditions, but this is not discussed. I feel that, given the large number of shortcomings, the results are not very meaningful, and they are probably mostly driven by the dominant role of the applied high-resolution digital elevation data.

In addition to these methodology deficiencies (and more of them are described in the detailed comments below), the paper is poorly written and organized, in some parts it is difficult to read and impossible to understand. A newer version of the paper will require a thorough proof-reading. I believe that the work is still too immature for publication. Here below are more detailed comments.

Detailed comments.

Introduction: it is currently a review of previous publications on the topic more than an introduction to the addressed issues. It should be synthesized, with focus on the issues that are addressed in the paper and on the gaps that the presented work will fill.

line 30-34: "Nevertheless, even if the literature for the smaller scales– that of the ripples, dunes, sastrugi and penitents – is usually distinct and scarcer, the principles equally apply to all the scales because the radiative transfers between faces are invariant by scale change" This is an example of tortuous sentence that need to be rephrased.

line 39: "…of the solar irradiance" It should be "of the direct solar irradiance".

line 51-52:" Arnold et al. (2006) also pointed out the role of the anisotropic reflectance of snow and ice, i.e. the fact that albedo is higher at higher solar zenith angles (Warren and Wiscombe, 1980)". This is a wrong explanation for the albedo dependence on the solar zenith angle. Anisotropy of snow reflectance has nothing to do with it (albedo is the integral of the directional reflectance over all azimuth angles). Albedo is higher at larger solar zenith angles because photons have larger probability of escaping to the atmosphere when they hit the snow at grazing angles. I have to say that this wrong explanation is also given in Arnold et al. (2006), who applied the correction factor of Lefebre et al (2003) to express the increase of snow albedo with increasing solar zenith angle, but wrongly attributed it to the nonisotropic reflectance properties of the snow.

line 53: "absorption enhancement is an additional effect…" You should specify that you refer to the absorption enhancement of solar radiation due to the orographic roughness. There are many other processes causing enhancement of absorbed energy…

line 61-63: "A simpler approach to account for multiple bounces is by assuming that the neighbouring faces are illuminated as if they were flat (Lenot et al., 2009; Olson et al., 2019). More importantly, the absorption enhancement is not uniform on the surface." This is an example of unclear and puzzling sentence: how do you possible account for multiple scattering between facets if the facets are not facing toward each other? I don't understand what you mean. Also, why the following sentence start with "More importantly"? More importantly than what?

line 76-77: "…finding deviations in surface solar fluxes on the order…" Deviations from what?

Figure 1: Please remove from the figure all the text that does not refer to the considered topographic effects and that is not referred to in the main text (name of models, energy fluxes, temperature lapse rate (text and diagram), wind speed and relative humidity).

Table 1: The title of the second column does not correspond to the content: you should replace "Spectral domain and illumination" with "Energy fluxes". Also, what is the difference between "self shadows" and "cast shadows"? They are not described in the text. And please replace "anisotropy of reflectance" with "solar zenith angle" effect (see above).

line 114-116: "The energy budget comprises (Arya, 1988): (i) the net radiation fluxes, which are split into the contributions of the short-wave radiation from 0.3 μm to 2 μm (SWnet) and the longwave radiation from 2 μm to 100 μm (LWnet)". I did not check the cited reference, but the correct wavelength intervals are 0.3 - 3μm for SWnet and 4-40 μm for LWnet. In fact, the downward longwave flux applied in the paper is measured by a CNR4 net radiometer, whose pyrgeometer senses the 4-42 μm wavelength window. Also, the irradiance in the 2-2.5 μm window is clearly part of the solar radiation spectrum, and not part of the thermal (longwave) radiation emitted by atmosphere and earth. By excluding the 2-2.5 μm window from the calculations of shortwave fluxes the authors significantly underestimate the surface net shortwave flux, as snow albedo is very low in this wavelength region. This is one of the major problems in this study.

Figure 2: it is too difficult to read. Please enlarge the font size and explain in the figure caption the meaning of TOM and of the terms in blue and grey.

Line 128-130: "The simulations are run in both direct and diffuse illumination conditions (noted with subscripts dir and diff), and the atmospheric effects (i.e. atmospheric attenuation) are neglected within the studied area (between the surface and TOM)." Here the text suggests that simulations are done in both clear and cloudy skies, which is clearly not the case, as all simulations are only done in clear-sky conditions and the model in developed for clear-sky conditions. This should by stated and clarified in the abstract, in the introduction, and here, when describing the modelling approach. Instead, I had to discover it only when the Landsat temperature scenes were described. The text could be improved by clarifying that the radiative transfer calculations are done separately for the direct and diffuse components of the clear-sky shortwave irradiance.

 Section 2.1.1: the major problem of this section is that equations 5 and 6 are not sufficiently explained. What is the meaning of $\alpha^i_{diff}$ and of the summation term in both equations? And the explanation given for the term $n^{(i)}_{hit,d,f}$ is not clear at all. The sentence "The RSRT model can indeed compute the number of times a photon has hit a given facet regardless of the albedo (and so of the wavelength), according to the bounce order of the photon (first reflection, second reflection, …)"

sounds odd: how can the number of scatterings of a photon on a facet be independent on the albedo of the facet? And how this is related to the derivation of $n_{hit,d,f}^{(i)}$? It is mentioned that some assumptions are made, but it is not explained what has been assumed. Finally, the explanation on how $I_{dir}$ and $I_{diff}$ are calculated is provided only in section 2.2., which in fact should be merged into 2.1.1.

Section 2.1.2: the main problem of this section is that the downward longwave flux is not corrected for variations in altitude over the 50 km$^2$ domain. I believe that this approximation is too crude, as it can cause an error in surface temperature of some °C when the differences in altitude are over 1000m (looking at the map, this difference seems to occur in the studied area). I recommend the authors to apply the correction, as done for instance by Arnold et al (2006). Another problem is the derivation of $LW_{u,scene-average}$: it is presented as a constant representing the average upwelling longwave flux from each facet. It is not explained how this quantity is calculated. The authors write that it is estimated according to Arnold et al (2006) so I went to read that article and found out that it is set to equal to the elevation-corrected air temperature in the surface grid mesh. Hence, it is not constant. The authors should explain in the paper how the variables are calculated, without requesting the reader to read the referred literature.

Section 2.1.4: this is the central and most problematic section. It should show how the surface energy budget is solved for Ts, but is totally unclear how equations 12, 13 and 14 are derived. It looks like that the Ts dependencies on air specific humidity and shortwave flux are ignored: is this the case? The extra equations in Appendix A3 are not of any help to understand the mathematical passages or the underlying assumptions, as they only show relationships between coefficients and not between Ts and the variables of the surface energy budgets.

Section 2.2: it should be merged to 2.1.1

Section 2.3: this section should describe the study area and the in-situ measurements, but it does not clarify which measurements were finally used. It is mentioned that meteorological and radiation data from FluxAlp station in Pre des Charmasses were used as input to the modelling chain, but which data were used from Col du Lautaret? And what is the elevation of these two stations? Automatic and manual measurements of SSA are mentioned, but it is not explained where and when they were measured (were they measured in each of the selected clear-sky days?). Since topography is the dominant feature addressed in the paper, it would be important to describe it more quantitatively: distribution of altitudes, distance between slopes, sizes of slopes. This quantitative information is also needed in the discussion, to explain the applicability of the method in other topographic environments.

line 282: "list in the appendix" should be "list in Appendix C"

Sections 3.2.1 and 3.2.2: In my opinion, validation of model simulations cannot be done with the same data used as input to the model. Hence, these two sections are meaningless and should be removed. The only aspect that could be saved is the comparison between modelled and observed shortwave radiation at FluxAlp, as in this case the simulation is independent from the observations. Actually, the comparison shows that the simulated net shortwave radiation is strongly underestimated, as expected because the simulation neglected the flux at wavelengths larger than 2000 nm (while the CNR4 pyranometers measure the radiation in the 300-3000 nm range).

Given the above considerations, I don't further comment the discussion and conclusion sections because I think they should be entirely rewritten once the listed methodological issues are solved.

---

## Author Comment (AC1)

**Answer to Anonymous Referee #1**

Referee comment on **tc-2021-180** : *Modelling surface temperature and radiation budget of snow-covered complex terrain* by Alvaro Robledano et al., The Cryosphere Discuss., https://doi.org/10.5194/tc-2021-180-RC1, 2021.

The reviewer's initial comments are written in black, and our answers are written in blue. The modifications and corrections in the paper are reported in *red* (the unchanged parts of the text are in *blue*). The line numbers, section numbers and figures correspond to those of the original manuscript.

The authors claim to estimate the LST and the energy budget of snow-covered complex terrains, in order to evaluate the significance of the different processes in influencing the spatial variations of the LST. The strategic analysis is interesting and significant for the scientific community. However, some issues remain to be discussed and some revisions are required before the manuscript could be accepted for publication. My specific comments are as follows.

The authors would like to thank Anonymous Referee #1 for the general analysis of the manuscript, as well as for the useful comments and suggestions, which we have taken into account. We wanted to note that a few changes in the modelling have been done based on the review by Anonymous Referee #2 and therefore almost all figures have been updated. Several parts of the manuscript have also been modified and rewritten. We show the new figures (and captions) at the end of the document, if not mentioned before. For specific details about these changes, we kindly refer the reviewer to author response AC2.

1. The authors claim to estimate the energy budget of the snow-covered complex terrains. However, this is not discussed in the results.

In this manuscript the focus is put on the snow surface temperature of complex terrains, and to reach it we need indeed to estimate the surface energy budget. The introduction was probably not very clear about this, so we have partly restructured it in order to highlight the importance of surface temperature.

Nevertheless, in Section 3.2.2 we have evaluated a diurnal cycle of all the simulated terms of the surface energy budget, in addition to surface temperature. This evaluation has been discussed in Section 4.2 (lines ~ 420 – 440), showing several strengths and weaknesses of the assumptions that have been made.

2. Why the double channel method (split window) is not examined by the authors?

Since the study goal is the modelling of the impact of topography on surface temperature, it is not a priority to evaluate it with different remote sensing algorithms. For this reason, we selected what

seemed to be the most reliable approach at the time of writing to provide the spatial variations of surface temperature.

As explained in Section 2.4 (line 265), stray light was observed on Landsat-8 thermal acquisitions (Montanaro et al., 2014), affecting Band 11. Even though several corrections have been applied afterwards (Gerace and Montanaro, 2017), we applied a method to retrieve surface temperature based on only one band, as suggested by other authors (e.g. Cristobal et al., 2018; He et al., 2019).

Indeed, the new Landsat Collection 2 Surface Temperature product relies on a single-channel approach, which supports our choice to rely on such an algorithm.

3. The discussion in the results is mostly qualitative. The analysis lacks statistical depth. The discussions include at places standard deviation, mean difference, etc., but not popular metrics such as the correlation coefficient and the RMSE. For example in Figure 8, 9 and 10.

We have updated Figures 8 and 9. The new Figure 8 includes a second panel with the RMSE and the correlation coefficient, that allows a more extensive discussion of the outliers (see comment 7d)

[Figure]

New *Figure 8: Comparison of the spatial variations of surface temperature between the simulations and the satellite observations for each date, computed considering the whole domain. On the left, mean bias and standard deviation of the differences. On the right, the RMSE and the correlation coefficient r.*

The new Figure 9 is changed. We compare now in the scatterplots the differences between each simulation and the reference simulation to better constrain the role of each topographic effect. We have added the correlation coefficient as suggested. By doing this, the evaluation of spatial variations with remote sensing observations is concentrated to Section 3.2.3, avoiding mixing the analysis of the topographic effects and the evaluation of the model. We have restructured the revised version of the manuscript to take this into account.

[Figure]

New *Figure 9: Impact of disabling a topographic effect on the simulated Ts on 18 February 2018. Every single panel corresponds to a disabled topographic effect, with respect to the reference simulation (REF) where all the effects are included. The marginal histograms show the distribution of surface temperature for each simulation as well as the observed Ts by the satellite (red) and the reference simulation (black).*

4. The authors mentioned and illustrated the effects of the topography on the estimated LST. However, they did not consider any literature on orographic corrections. For example as follows which is replicable for LST in a similar manner:

Bento et al., 2017: https://doi.org/10.3390/rs9010038

Varade and Dikshit, 2019: https://doi.org/10.1029/2018WR023806

The study is focused on modelling. While we use LST retrieved from Landsat-8 to compare with the spatial variations predicted by the model, improving LST retrieval algorithms or comparing different satellite products is out of the scope of our study (and expertise). It would not benefit the main goal of this work. We clarify this aspect in the revised version (e.g. change in Figure 9). However, we acknowledge that the remote sensing topic is related and likely of interest for the readers interested by this study. For this we propose to add the reference suggested by the reviewer in the comment 5 in the Section 4.1 (line 409), as follows:

*A possible future improvement would be to include a land mask to set a particular emissivity value for each pixel depending on the presence of snow, rocks, grass, etc. This is normally achieved by means of NDVI-based classifications (Li et al., 2013), that can be adapted to snow-covered complex terrains with methods that rely on the snow cover area (Varade and Dikshit, 2020).*

5. The authors mentioned the limitations of the NDVI thresholds method for the estimation of emissivity. The authors may explore the following alternative:

Varade and Dikshit, 2020: DOI: 10.1080/10106049.2018.1520928

This alternative has been added as suggested (see above).

Further, the authors missed the influence of the vegetation or the forest cover in their analysis, which is significant on the LST and the atmospheric water vapor content.

The influence of vegetation is neglected here as its presence is very limited in the study area. The following image is a screenshot of a 360-degree webcam operating at the Col du Lautaret mountain pass, corresponding to the 18th February 2018 (one of the selected Landsat-8 acquisition dates):

[Figure]

For clarity, we have added the following (line 239):

*The predominant orientation is S-SW, followed by N-NE facing terrain, and the slope varies mainly between 15° and 40°. Protruding vegetation is rare in the study area in winter and is neglected here. We assume that the snow cover is 100%.*

6. Since, the comparison is made against the Landsat-8 derived LST, it is imperative that the used reference product is at the most best quality. I would recommend the authors to calibrate this product from a series of ground station data if available.

This study uses Landsat-8 to compare the spatial variations of LST predicted by the model, but does not rely on the absolute accuracy of the product. The bias identified at the single meteorological station available in our domain could be used to apply an offset to the satellite product but there would be no gain for the spatial variations. For the sake of simplicity and to keep the manuscript focused on the modelling aspect, we prefer to keep the remote sensing part as simple as possible.

In more details:
The Landsat Collection 2 Surface Temperature product was not available when we first started this work. However, when it was made available, we assessed its accuracy at the in-situ station and compared it with the already implemented retrieval method. Figure 5a shows that the reference

product is less accurate at this particular point, with a bias of -3.5°C (RMSE: 4.0°C). The applied single-channel algorithm seems to be more accurate at this place, with a bias of -1.3°C (RMSE: 2.0°C). On the other hand, as stated in Section 4.1 (line 413), the differences across the study area between the applied method and the official product are of 0.3 °C (median of standard deviations). Considering that both products are virtually equivalent regarding the spatial variations (i.e. the main goal of the study) and the better accuracy at the measurement station, we decided to keep the single-channel algorithm proposed by Cristobal et al. (2018) as our reference product.

7. Comments regarding the write-up.

a) The language of the manuscript is extremely poor. It is difficult to understand because of the poor language used. The following checks are required by the authors

     i. Missing punctuations. Example- Line 1,5 in abstract.

     ii. Grammatical mistakes, usage of incorrect articles.

     iii. Usage of appropriate words. For example, Line 28, "Terrain tilt", I believe should be "Terrain orientation" or "Terrain slope" . The sentence is very difficult to understand and there are several such sentences in the manuscript.
     Another example, Line 214 it should be "quadratic". And so on.

The manuscript will be revised beforehand by an English native speaker.
Line 28: "*terrain tilt*" has been modified in the text by *"terrain slope and orientation"*.

Line 214: the general form of a quartic equation is:
$ax^4 + bx^3 + cx^2 + dx + e = 0$

while the general form of a quadratic equation is:
$ax^2 + bx + c = 0$

So in our case, **quartic** equation is the appropriate word, there is no error.

b) Figure 3, instead of showing the chart in the left image, the authors can show the slopes and their directions using directional gradient filters applied on the DEM.

The Figure has been updated as suggested.

[Figure]

New Figure 3: *Location of the study area, around the Col du Lautaret alpine site. The blue rectangle in (a) represents the* extension of the study area, shown in (b). The domain is generated *from the RGE ALTI®Version 2.0 Digital Elevation Model (DEM) provided by IGN France at a spatial resolution of 5m, and resampled to 10m for this study.* Slope angle is represented by the intensity of the color.

c) Abbreviations/symbols needs to be defined in Figures, For example in Figure 1 and 2. In some cases, the definitions of these come after several paragraphs or in other sections.

Figures and captions have been updated following the comments of both reviewers. We have removed part of the text in Figure 1 to make it more clear:

[Figure]

Figure 2 has been updated by enlarging the font size and modifying the caption:

[Figure]

*Figure 2: Flowchart of the modelling chain to estimate snow surface temperature. TOM (top of mountains) is the horizontal above the highest point in the study area. The involved models are in green, the terms of the surface energy budget are in orange, the needed inputs are in blue and the topographic effects are in grey. The red dashes lines indicate the two-step iterative process to compute the downward LW flux.*

d) The discussion of the some of the Figures and corresponding results is not sufficient. For example, in Figure 8, bulk of the points are between σ of ~1-3 o C, Hoverever, some outliers are also observed. These are not discussed in the manuscript. Any particular reasons for this.

Figure 8 has been updated following the major revisions suggested by Anonymous Reviewer #2 and the previous comments of the reviewer. Some outliers are indeed observed, and a plausible explanation is that in the modelling chain we consider a totally snow-covered area. However this is potentially not the case for several acquisition dates, in particular in early winter and early spring. We have modified the discussion of the outliers (~ line 340) as follows:

*The simulations are slightly colder in general, with a bias principally between -3°C and 1°C. The standard deviation of the differences varies mostly between 1 and 3°C (2 and 4°C for the RMSE). Some outliers are observed, and in particular the simulation that shows the highest differences (both standard deviation and RMSE) correspond to an acquisition from late March. Such differences could be explained with an early onset of snowmelt (snow patches in the lowest areas) due to mild temperatures, a particular situation that breaks the assumptions in the model (e.g.*

*100% snow cover). The shallow snowpack in early winter (probable patches of bare soil) can lead to a similar situation, where the bare soil temperature would be certainly different than that of snow-covered terrain. This could explain the lowest correlation value of the dataset which corresponds to an acquisition from early December.*

e) Figure 7, it would be interesting to see how a downscaled Landsat-8 LST would fare against the results from the proposed methodology.

We kindly refer to our answers to comments 2, 4 and 6.

**References:**

Cristóbal, J., Jiménez-Muñoz, J., Prakash, A., Mattar, C., Skoković, D., and Sobrino, J.: An Improved Single-Channel Method to Retrieve Land Surface Temperature from the Landsat-8 Thermal Band, Remote Sensing, 10, 431, https://doi.org/10.3390/rs10030431, 2018.

Gerace, A., and M. Montanaro. 2017.: Derivation and Validation of the Stray Light Correction Algorithm for the Thermal Infrared Sensor Onboard Landsat 8, Remote Sensing of Environment, 191 (Supplement C), 246–257. https://doi.org/10.1016/j.rse.2017.01.029, 2017.

He, J., Zhao, W., Li, A., Wen, F., & Yu, D.: The impact of the terrain effect on land surface temperature variation based on Landsat-8 observations in mountainous areas. International Journal of Remote Sensing, *40*(5–6), 1808–1827,https://doi.org/10.1080/01431161.2018.1466082, 2019.

Varade, D. & Dikshit, O.: Assessment of winter season land surface temperature in the Himalayan regions around the Kullu area in India using landsat-8 data, Geocarto International, 35:6, 641-662, https://doi.org/10.1080/10106049.2018.1520928, 2020.

**Extra figures:**

[Figure]

New Figure 5

[Figure]

New Figure 6: *Simulation of the surface fluxes (top) and snow surface temperature (bottom) at the FluxAlp station for a ~36h long time series starting 10 March 2016. The radiative fluxes are compared to in situ measurements. All times are in UTC.*

[Figure]

New Figure 7

[Figure]

New Figure 10

[Figure]

New Figure 11

[Figure]

New Figure 12

---

## Author Comment (AC2)

**Answer to Anonymous Referee #2**

Referee comment on **tc-2021-180** : *Modelling surface temperature and radiation budget of snow-covered complex terrain* by Alvaro Robledano et al., The Cryosphere Discuss., https://doi.org/10.5194/tc-2021-180-RC2, 2021.

The reviewer's initial comments are written in black, and our answers are written in blue. The modifications and corrections in the paper are reported in *red* (the unchanged parts of the text are in *blue*). The line numbers, section numbers and figures correspond to those of the original manuscript.

The topic of the manuscript is very interesting and challenging. The authors claim that their developed modelling procedure, which involves several steps and the use of different schemes, enables the calculation of the surface temperature and surface energy budget over snow-covered mountain areas at high spatial resolution (10m). Although some of the results seem encouraging, the method is not clearly described, and there are macroscopic shortcomings in the modelling, the biggest ones being that the applied downward longwave radiation is not corrected for variations in altitude, and that the equation to extract the surface temperature from the surface energy balance equation is totally obscure and seems rather arbitrary (looks like that the Ts dependency on air specific humidity and shortwave flux are ignored?). Also, the solar infrared flux for wavelengths longer than 2000 nm is neglected, without explaining neither the reason for the neglection nor the implications of this neglection in the results (actually, this is the case also for other applied approximations). Finally, the model seems applicable only for clear-sky conditions, but this is not discussed. I feel that, given the large number of shortcomings, the results are not very meaningful, and they are probably mostly driven by the dominant role of the applied high-resolution digital elevation data. In addition to these methodology deficiencies (and more of them are described in the detailed comments below), the paper is poorly written and organized, in some parts it is difficult to read and impossible to understand. A newer version of the paper will require a thorough proof-reading. I believe that the work is still too immature for publication.

The authors would like to thank the Anonymous Reviewer #2 for the useful comments and suggestions. We have taken all of them into account in order to improve this work and make it suitable for publication. The errors pointed in the modelling by the reviewer have been corrected and led to improved results. The method has been explained in a clearer way and several parts of the manuscript have been restructured and rewritten. The new manuscript will be revised beforehand by an English native speaker.

Here below are more detailed comments.

Introduction: it is currently a review of previous publications on the topic more than an introduction to the addressed issues. It should be synthesized, with focus on the issues that are addressed in the paper and on the gaps that the presented work will fill.

As the study focuses on specific topographic effects, we think that presenting the literature on this topic is a possible way to introduce the addressed issues. This way shows the different levels of complexity, what are the current limitations and how this work can provide new solutions. Several sentences have been however rephrased and several paragraphs have been synthesized in order to highlight the main goal of the study.

line 30-34: "Nevertheless, even if the literature for the smaller scales– that of the ripples, dunes, sastrugi and penitents – is usually distinct and scarcer, the principles equally apply to all the scales because the radiative transfers between faces are invariant by scale change" This is an example of tortuous sentence that need to be rephrased.

The sentence has been entirely removed.

line 39: "...of the solar irradiance" It should be "of the direct solar irradiance".

Done.

line 51-52:" Arnold et al. (2006) also pointed out the role of the anisotropic reflectance of snow and ice, i.e. the fact that albedo is higher at higher solar zenith angles (Warren and Wiscombe, 1980)". This is a wrong explanation for the albedo dependence on the solar zenith angle. Anisotropy of snow reflectance has nothing to do with it (albedo is the integral of the directional reflectance over all azimuth angles). Albedo is higher at larger solar zenith angles because photons have larger probability of escaping to the atmosphere when they hit the snow at grazing angles. I have to say that this wrong explanation is also given in Arnold et al. (2006), who applied the correction factor of Lefebre et al (2003) to express the increase of snow albedo with increasing solar zenith angle, but wrongly attributed it to the nonisotropic reflectance properties of the snow.

Our explanation was not correct, so the sentence will be removed because it is not the main topic of this study. Many studies have already addressed this topic in the past.

line 53: "absorption enhancement is an additional effect..." You should specify that you refer to the absorption enhancement of solar radiation due to the orographic roughness. There are many other processes causing enhancement of absorbed energy…

It has been modified as:

*Absorption enhancement due to multiple scattering within the topography is an additional effect...*

line 61-63: "A simpler approach to account for multiple bounces is by assuming that the neighbouring faces are illuminated as if they were flat (Lenot et al., 2009; Olson et al., 2019). More importantly, the absorption enhancement is not uniform on the surface." This is an example of unclear and puzzling sentence: how do you possible account for multiple scattering between facets if the facets are not facing toward each other? I don't understand what you mean. Also, why the following sentence start with "More importantly"? More importantly than what?

It is indeed unclear. The approximation is described in two studies:

Lenot et al., 2009: "Over rugged terrain, both irradiance at ground level and atmospheric albedo vary. The coupling irradiance can be estimated with reasonable accuracy using a Monte-Carlo code, but this approach is too time consuming for the present discussion. As there is no alternative way to calculate it accurately, a rough estimate of it, based on flat terrain, is used: (Eq. 7) "

Olson et al., 2019: "Rather than attempting to accurately determine the albedo and scattering direction of radiation from all nearby terrain, solar radiation models often use a single value to represent the albedo of surrounding terrain and multiply the sky-view factor by the amount of direct and diffuse irradiance arriving on a flat plane (Eq. 3)"

We have rephrased our summary of this approach:

*A simpler approach to account for multiple bounces is to add a mean contribution coming from neighbouring slopes, by assuming that they are illuminated as if they were flat (Lenot et al., 2009; Olson et al., 2019). This contribution requires a value representing the effectif albedo of neighbouring terrain and is modulated by the sky view factor of the slope. It is important to note that the absorption enhancement is not uniform, it is usually stronger in the valleys (trapping effect) than near the summits of the topography (Lliboutry, 1954).*

line 76-77: "...finding deviations in surface solar fluxes on the order..." Deviations from what?

We have modified the sentence as follows:

*Lee et al. (2013) used similar methods to show the impact of including the topography on surface radiation budget over the whole Tibetan Plateau. They found differences of ~14 Wm$^{-2}$ with respect to a flat surface calculation.*

Figure 1: Please remove from the figure all the text that does not refer to the considered topographic effects and that is not referred to in the main text (name of models, energy fluxes, temperature lapse rate (text and diagram), wind speed and relative humidity).

The name of models are now removed, as well as the relative humidity. The rest is referred to in the main text when describing all the topographic effects.

[Figure]

Table 1: The title of the second column does not correspond to the content: you should replace "Spectral domain and illumination" with "Energy fluxes". Also, what is the difference between "self shadows" and "cast shadows"? They are not described in the text. And please replace "anisotropy of reflectance" with "solar zenith angle" effect (see above).

We have replaced "*Spectral domain and illumination*" with "*Energy flux*" as suggested. "*Anisotropy of reflectance*" has been removed as is not the main topic of this study (see above).

Cast shadows are the shadows that are cast by an object (topography in this case) when it occludes the light source. Self shadows are in the occluding object itself, and they are related to the local solar zenith angle (line 40 – *it happens when a face completely turns away from the sun*). The manuscript has been modified as follows:

*(line 38) The first effect is the combination of the shadowing from local horizons (cast shadows) and the modulation of the direct solar irradiance depending on the face slope and aspect relative to the sun's position. This modulation depends on the local solar zenith angle (SZA). Self-shadowing occurs when the face completely turns away from the sun (local SZA < 0° or > 90°).*

The order of the first two rows of Table 1 has been inverted and now reads:

*Cast shadows*

*Variations of the local solar zenith angle (self shadows)*

line 114-116: "The energy budget comprises (Arya, 1988): (i) the net radiation fluxes, which are split into the contributions of the short-wave radiation from 0.3 μm to 2 μm (SWnet) and the longwave radiation from 2 μm to 100 μm (LWnet)". I did not check the cited reference, but the correct wavelength intervals are 0.3 - 3μm for SWnet and 4-40 μm for LWnet. In fact, the downward longwave flux applied in the paper is measured by a CNR4 net radiometer, whose pyrgeometer senses the 4-42 μm wavelength window. Also, the irradiance in the 2-2.5 μm window is clearly part of the solar radiation spectrum, and not part of the thermal (longwave) radiation emitted by atmosphere and earth. By excluding the 2-2.5 μm window from the calculations of shortwave fluxes the authors significantly underestimate the surface net shortwave flux, as snow albedo is very low in this wavelength region. This is one of the major problems in this study.

We thank the reviewer for raising this important issue. We had stopped the SW simulations at 2μm considering that no spatial variations would be induced by multiple scattering in the upper wavelengths, and because of the necessity to limit the computational cost of applying Eqs. (5),(6) for every facet at every wavelength. However, the impact of this choice turns out to be significant as noted by the reviewer.

We have now extended the SW simulations to 4.2μm, with the same 3nm spectral resolution. This choice is motivated by the spectral range of the CNR4 net radiometer (0.3 – 2.8μm in the SW; 4.5 – 42μm in the LW). As we apply the downward LW flux measured by the pyrgeometer, we have decided to cover (almost) the rest of the solar radiation spectrum with the SW simulations. The following figure shows the impact of this extension on the simulated surface temperature, downward SW flux and net SW flux at noon:

[Figure]

The underestimation of the surface $SW_{net}$ flux in the first version of the manuscript was indeed significant (~ 30 $Wm^{-2}$), and the impact on surface temperature was an offset of about 1°C. As a consequence, all the simulations have been rerun and the related figures have been reproduced to account for these new SW simulation. In the specific case of Figure 6 (top), for a fairly comparison between the simulated and the measured $SW_{net}$ flux, additional SW calculations have been done in the same spectral range of the pyranometer (0.3 – 2.8μm). The new figures appear in this response when needed, and the rest are at the end of the document.

Line 114-116 has been modified as follows:

*The energy budget comprises (Arya, 1988): (i) the net radiation fluxes, which are split into the contributions of the short-wave radiation from 0.3µm to 4µm (Swnet) and the long-wave radiation from 3µm to 100µm (Lwnet);*

Figure 2: it is too difficult to read. Please enlarge the font size and explain in the figure caption the meaning of TOM and of the terms in blue and grey.

The figure has been updated as suggested:

[Figure]

*Figure 2: Flowchart of the modelling chain to estimate snow surface temperature. TOM (top of mountains) is the horizontal above the highest point in the study area. The involved models are in green, the terms of the surface energy budget are in orange, the needed inputs are in blue and the topographic effects are in grey. The red dashes lines indicate the two-step iterative process to compute the downward LW flux.*

Line 128-130: "The simulations are run in both direct and diffuse illumination conditions (noted with subscripts dir and diff), and the atmospheric effects (i.e. atmospheric attenuation) are neglected within the studied area (between the surface and TOM)." Here the text suggests that simulations are done in both clear and cloudy skies, which is clearly not the case, as all simulations are only done in clear-sky conditions and the model in developed for clear-sky conditions. This should by stated and clarified in the abstract, in the introduction, and here, when describing the modelling approach. Instead, I had to discover it only when the Landsat temperature scenes were described. The text could be improved by clarifying that the radiative transfer calculations are done separately for the direct and diffuse components of the clear-sky shortwave irradiance.

The study is indeed interested in clear-sky conditions because part of the topographic effects would vanish under fully diffuse illumination. We forgot to state this important point.

Note that the methodology and the models themselves are general enough and can be run with clouds, but the application in the study is limited to clear-sky conditions. We have improved the text to make this clear from the beginning (abstract and introduction).

Line 128-130 has been modified as suggested:

*The simulations are done separately for the direct and diffuse components of the short-wave irradiance (noted with subscripts dir and diff). Moreover, the atmospheric effects (i.e. atmospheric attenuation, clouds) are neglected within the studied domain (between the surface and TOM)*

Section 2.1.1: the major problem of this section is that equations 5 and 6 are not sufficiently explained. What is the meaning of $\alpha^i_{diff}$ and of the summation term in both equations? And the explanation given for the term $n^{(i)}_{hit,d,f}$ is not clear at all. The sentence "The RSRT model can indeed compute the number of times a photon has hit a given facet regardless of the albedo (and so of the wavelength), according to the bounce order of the photon (first reflection, second reflection, ...)"sounds odd: how can the number of scatterings of a photon on a facet be independent on the albedo of the facet? And how this is related to the derivation of $n^{(i)}_{hit,d,f}$ ? It is mentioned that some assumptions are made, but it is not explained what has been assumed. Finally, the explanation on how $I_{dir}$ and $I_{diff}$ are calculated is provided only in section 2.2., which in fact should be merged into 2.1.1.

The explanation was indeed not clear enough, in part because there was a typo in Eq. (5), that now reads:

$$A_{\text{dir, f}}(\lambda, \theta_s) = (1 - \alpha_{\text{dir}}(\lambda, \theta_s))\, n^{(0)}_{\text{hit, dir, f}} + (1 - \alpha_{\text{diff}}(\lambda))\, \alpha_{\text{dir}}(\lambda, \theta_s) \sum_{i=1}^{i=n_{\max}} \alpha^{i-1}_{\text{diff}}(\lambda)\, n^{(i)}_{\text{hit, dir, f}} \tag{5}$$

$$A_{\text{diff, f}}(\lambda) = (1 - \alpha_{\text{diff}}(\lambda)) \sum_{i=0}^{i=n_{\max}} \alpha^{i}_{\text{diff}}(\lambda)\, n^{(i)}_{\text{hit, diff, f}} \tag{6}$$

The RSRT model computes the number of times a photon has hit a given facet, according to the bounce order of the photon (first reflection, second reflection...) and assuming no absorption. This number is pure geometrical calculation of trajectory and is independent of facet albedo. This number is useful to deduce the albedo with Eqs. (5) and (6) depending on the illumination. For each facet f of the modelled surface, $n^{(i)}_{hit,d,f}$ corresponds to the proportion of photons that hit the facet on their $i^{th}$ reflection (starting from i = 0). For a facet f, $n^{(0)}_{hit,d,f}$ represents the proportion of photons that have directly hit the facet at their first reflection, $n^{(1)}_{hit,d,f}$ at their second reflection, etc.

In order to compute the absorption coefficients we take advantage from this and from the assumption that the area has the same snow properties everywhere and that each reflection is Lambertian. So, in the simulation with direct illumination, each facet will receive direct illumination only from photons at their first reflection, while the "multiple bouncing" effect is taken into account as diffuse illumination from photons at their second, third, ...reflection. The summation term in Eqs. (5) and (6) is, indeed, the multiple bounce contribution. The same applies to the simulations with diffuse illumination but in a simpler way, as every bounce comes from diffuse illumination.

$\alpha^i_{diff}$ is the snow spectral albedo in diffuse illumination, raised to the i[th] power.

For clarity, we have partly rewritten and restructured the section, as well as merged the Section 2.2 into this section as suggested.

Section 2.1.2: the main problem of this section is that the downward longwave flux is not corrected for variations in altitude over the 50 km$^2$ domain. I believe that this approximation is too crude, as it can cause an error in surface temperature of some °C when the differences in altitude are over 1000m (looking at the map, this difference seems to occur in the studied area). I recommend the authors to apply the correction, as done for instance by Arnold et al (2006).

We thank the reviewer for this relevant suggestion. The variations in altitude within the domain are indeed over 1000m (in particular between 1640 m.a.s.l. and 3220 m.a.s.l.), so an altitudinal correction to the downward LW flux is justified. We now apply a simple correction, as done by Arnold et al. (2006). The correction uses the Stefan-Boltzmann equation and the measured downward LW flux to derive an "effective emissive temperature" of the sky, which is then corrected for variations in altitude as we do with the air temperature. The elevation-corrected $LW_d$ for each facet of the mesh is then applied in the modelling chain. The changes in the manuscript are specified in the following comment as they are related.

For a selected simulation (18 February 2018), the measured $LW_d$ is 204 Wm$^{-2}$, the median value over the whole domain is 198 Wm$^{-2}$ and the extreme values (min, max) are 180 and 213 Wm$^{-2}$ respectively. These values are similar (same order of magnitude) to those found by other authors (Greuell et al., 1997; Iziomon et al., 2003). To quantify the impact of this correction on snow surface temperature, we have updated the Figures 9 and 10 with an additional "topographic effect", named "No $LW_d$ correction":

[Figure]

*New Figure 9* (please note that the scatterplots have also been modified to accomodate the Anonymous Reviewer #1 suggestions)*: Impact of disabling a topographic effect on the simulated Ts on 18 February 2018. Every single panel corresponds to a disabled topographic effect, with*

[Figure]

*respect to the reference simulation (REF) where all the effects are included. The marginal histograms show the distribution of surface temperature for each simulation as well as the observed Ts by the satellite (red) and the reference simulation (black)*

*New Figure 10*

The impact on the spatial variations of surface temperature is however not very significant, less important than the rest of considered topographic effects. We have added in the new manuscript an interpretation and a discussion on this new effect in Sections 3 and 4.

Another problem is the derivation of $LW_{u,scene-average}$ : it is presented as a constant representing the average upwelling longwave flux from each facet. It is not explained how this quantity is calculated. The authors write that it is estimated according to Arnold et al (2006) so I went to read that article and found out that it is set to equal to the elevation-corrected air temperature in the surface grid mesh. Hence, it is not constant. The authors should explain in the paper how the variables are calculated, without requesting the reader to read the referred literature.

There was indeed a mistake in the text as correctly raised by the reviewer, caused by a misinterpretation of the literature by the authors. The variable $LW_{u, scene-averaged}$ is the upwelling longwave flux calculated with the Stefan-Boltzmann equation from the average surface temperature of the whole domain. To be clearer, this is a three-step process: we first derive the snow surface temperature for each facet without accounting for the thermal emission of surrounding terrain. We then derive the thermal contribution ($LW_{u, scene-averaged}$) with the average Ts of the domain and finally recalculate the surface temperature of each facet with Eq. (7).

The whole section has been rewritten and restructured as follows:

*To compute the incident LW flux from the atmosphere on each facet, we first apply a local correction to the observed downward LW flux (noted $LW_{d, obs}$), measured in a single point of the domain (Sect. 2.2). This correction takes into account the LW variations in altitude over the whole domain, using the same approach as in Arnold et al. (2006). The correction consists in deriving an "effective emissive temperature" of the sky (noted $T_{sky, obs}$) from the measured LW flux with the Stefan-Boltzmann equation. This temperature is corrected for changes in elevation by introducing the lapse rate $\Gamma$:*

$$LW_{d, obs} = \sigma \ T^4_{sky, obs}$$

$$T_{sky, f} = \ T_{sky, obs} + \Gamma \ (z_f - z_{obs})$$

*with $\sigma$ the Stefan-Boltzmann constant and where $T_{sky, f}$ is the effective emissive temperature of the atmosphere above each facet. We choose $\Gamma = -6.5 °C \ km^{-1}$, the environmental lapse rate as defined in the International Standard Atmosphere. The downwelling LW flux incident on each facet ($LW_{d, f}$) is then recalculated with the Stefan-Boltzmann equation:*

$$LW_{d, f} = \sigma \ T^4_{sky, f}$$

*In complex terrain facets not only receive radiation from the atmosphere but also from the surrounding slopes. Computing this contribution requires the facet surface temperature, which is precisely unknown. We proceed by iteration in two steps: in the first step, we neglect the thermal emission from the surrounding facets. This leads to a first estimate of the surface temperature (Sect. 2.1.4), that is then used in the second step to account for the emission of surrounding terrain. The average upwelling LW flux from each facet ($LW_{u,scene-average}$) is computed from the average surface temperature of the whole domain with the Stefan-Boltzmann equation. This thermal emission is a constant, we neglect the possible variations of temperature around each facet. The updated downwelling LW flux on each facet is eventually calculated with:*

$$LW_{d,f, updated} = V_f \ LW_{d,f} + (1 - V_f) \ LW_{u, scene-average}$$

*where $V_f$ is the sky-view factor calculated with RSRT. $V_f$ is different for each facet: those in the valley receive more LW radiation from the surrounding slopes than facets at the summits of the domain. The sky-view factor is indeed equal to the proportion of the launched photons hitting a facet on their first bounce in diffuse illumination, namely $V_f = n^{(0)}_{hit, diff, f}$.*

*The upwelling long-wave radiation, $LW_{u, f}$ is determined by the Stefan-Boltzmann law:*

$$LW_{u, f} = \varepsilon \sigma \ T^4_s + (1 - \varepsilon) \ LW_{d, f}$$

*with snow emissivity $\varepsilon = 0.98$ and $T_s$ the snow surface temperature of the facet.*

Note that Section 2.1.3 has also been modified to accomodate the earlier introduction of the lapse rate.

Section 2.1.4: this is the central and most problematic section. It should show how the surface energy budget is solved for Ts, but is totally unclear how equations 12, 13 and 14 are derived. It looks like that the Ts dependencies on air specific humidity and shortwave flux are ignored: is this the case? The extra equations in Appendix A3 are not of any help to understand the mathematical passages or the underlying assumptions, as they only show relationships between coefficients and not between Ts and the variables of the surface energy budgets.

We acknowledge that the text was not clear enough to understand how the surface energy budget is solved. Starting from Eq. (1) with explicit Ts dependencies:

$SW_{net,f} + LW_{d,f,updated} - LW_{u,f} (\textbf{\textit{Ts}}^4) + H_f (\textbf{\textit{Ts}}) + L_f (qsat(\textbf{\textit{Ts}})) = 0$

In order to avoid solving this non-linear equation for millions of facets (which would imply an enormous computational cost), the non-linear term in Ts (air specific humidity) is linearized about Tair, as in Essery (2004) or Best et al. (2004).

Eqs. (12) and (13) explain the linearization, and so the surface energy budget equation can eventually be expressed as a quartic equation of the form:

$a\ Ts^4 + d\ Ts + e = 0$

whose general solution requires a simple change of variable to transform the quartic into a depressed quartic:

$Ts^4 + pTs^2 + q\ Ts + r = 0$

that can be solved by means of the Ferrari solution (Neumark, 1965). It consists in adapting the equation to present it as a difference of two squares, which eventually leads to a resolvent cubic that is then solved, yielding:

$$Ts = -S + \frac{1}{2}\sqrt{-4S^2 + \frac{q}{S}}$$

that has already been developed in the Appendix A3. We have rewritten the section to add more details, in particular the linearization step to derive the quartic equation for Ts and the derivation of the general solution in the Appendix.

Section 2.2: it should be merged to 2.1.1

It has been merged.

Section 2.3: this section should describe the study area and the in-situ measurements, but it does not clarify which measurements were finally used. It is mentioned that meteorological and radiation data from FluxAlp station in Pre des Charmasses were used as input to the modelling chain, but which data were used from Col du Lautaret? And what is the elevation of these two stations?

The Col du Lautaret is a mountain pass and is used in the text as a geographic reference. All the in situ measurements come from the unique FluxAlp station in Pré des Charmasses site. The station is located at 2052 m.a.s.l, while the mountain pass is a few meters higher. To clarify, the "Col du Lautaret" marker in Figure 3 has been removed and the introductory sentences of the section have been modified as follows:

*(line 235) Figure 3 shows the study area. It is located around the Lautaret mountain pass in the French Alps (45.0°N, 6.4°E)*

*[...]*

*(line 244) The study area also includes the measurement station FluxAlp (45.0413°N, 6.4106°E),  located at 2052 m.a.s.l.*

Automatic and manual measurements of SSA are mentioned, but it is not explained where and when they were measured (were they measured in each of the selected clear-sky days?).

Manual measurements of SSA were collected occasionally during two consecutive winter seasons (2016/2017 and 2017/2018 – Tuzet et al. (2020)) as mentioned in line 253. They were collected a short distance away (few meters) from the automatic measurement station (FluxAlp). There were no automatic measurements of SSA.
Among the selected clear-sky days, 4 of them had an accompanying SSA measurement (2 February 2018, 18 February 2018, 27 February 2018 and 22 March 2018). For the rest of selected clear-sky days we assume a standard value for SSA (20 $m^2kg^{-1}$). The paragraph has been modified to address these points:

*(line 253)  Manual measurements of surface snow SSA for two consecutive winter seasons (2016/2017 and 2017/2018) have been collected occasionally (Tuzet et al., 2020), a few meters around the FluxAlp station.*

*(line 283) The acquisition time of Landsat-8 observations (10h17 or 10h23 UTC depending on the scene) and in situ measurements (10h30 UTC, averaged over the previous 30min) are considered to be equivalent. Among the selected clear-sky days, four of them have an accompanying manual measurement of SSA (Tuzet et al., 2020). These are 2 February 2018, 18 February 2018, 27 February 2018 and 22 March 2018, with snow SSA of 47, 45, 53 and 32 $m^2kg^{-1}$, respectively.*

Since topography is the dominant feature addressed in the paper, it would be important to describe it more quantitatively: distribution of altitudes, distance between slopes, sizes of slopes. This quantitative information is also needed in the discussion, to explain the applicability of the method in other topographic environments.

The figure representing the study area has been modified following the Anonymous Reviewer #1's comments. We have added to the text the following:

*(line 235) This area is of interest to study surface temperature, as it features both north and south-facing slopes that are spaced by a few hundreds of meters. It also contains smaller-scale rugged terrain covering the rest of orientations and promoting re-illumination. The size of the area is ~ 50 km², and the range of altitudes spans from 1640 to 3220 m.a.s.l. The predominant orientation is S-SW, followed by N-NE facing terrain, and the slope varies mainly between 15° and 40°. Protruding vegetation is rare in the study area in winter and is neglected here. We assume that the snow cover is 100%.*

line 282: "list in the appendix" should be "list in Appendix C"

Done.

Sections 3.2.1 and 3.2.2: In my opinion, validation of model simulations cannot be done with the same data used as input to the model. Hence, these two sections are meaningless and should be removed. The only aspect that could be saved is the comparison between modelled and observed shortwave radiation at FluxAlp, as in this case the simulation is independent from the observations. Actually, the comparison shows that the simulated net shortwave radiation is strongly underestimated, as expected because the simulation neglected the flux at wavelengths larger than 2000 nm (while the CNR4 pyranometers measure the radiation in the 300-3000 nm range).

The evaluation of the model is done with data independent from the inputs, which are the following:

- Incoming solar irradiance: output of SBDART model and involved in the SW calculation
- Snow SSA and roughness length: involved in the SW and turbulent heat fluxes calculations, respectively.
- Downward LW flux
- Wind speed, air temperature and relative humidity: involved in the turbulent heat fluxes calculation.

The SW simulation is indeed independent from the observation at the FluxAlp station, as mentioned by the reviewer. For the specific comparison in Figure 6 (top), the underestimation is now less important as we have extended the SW simulations to 2.8 µm, in order to cover the spectral range of the CNR4 pyranometer (0.3 – 2.8 µm).

The computation of the downwelling long-wave flux over each facet is an iterative process, which includes the contribution of the thermal emission from surrounding terrain. As mentioned above, this thermal emission depends on an initial estimation of the surface temperature. So, even if a measured value is needed as input, the eventually estimated $LW_d$ (and therefore $LW_u$ and $LW_{net}$ ) over each facet depends on the simulation results, and not only on the observations.

With respect to the turbulent heat fluxes, the comparison was not appropriate. In the updated Figure 6 (top), only the simulated turbulent fluxes are shown, and no comparison is made in the text.

[Figure]

New Figure 6: *Simulation of the surface fluxes (top) and snow surface temperature (bottom) at the FluxAlp station for a ~36h long time series starting 10 March 2016. The radiative fluxes are compared to in situ measurements. All times are in UTC.*

Given the above considerations, I don't further comment the discussion and conclusion sections because I think they should be entirely rewritten once the listed methodological issues are solved.

The authors understand the difficulty to continue the review with so many doubts about the methods. We hope to have clarified the manuscript. The new results are improved but do not significantly change our initial findings and conclusions.

References:

Arnold, N. S., Rees, W. G., Hodson, A. J., and Kohler, J.: Topographic controls on the surface energy balance of a high Arctic valley glacier, Journal of Geophysical Research, 111, https://doi.org/10.1029/2005jf000426, 2006.

Best, M. J., Beljaars, A., Polcher, J., & Viterbo, P.: A Proposed Structure for Coupling Tiled Surfaces with the Planetary Boundary Layer. *Journal of Hydrometeorology*, *5*(6), 1271–1278. https://doi.org/10.1175/JHM-382.1, 2004.

Essery, R.: Parameter sensitivity in simulations of snowmelt. *Journal of Geophysical Research*, *109*(D20), D20111. https://doi.org/10.1029/2004JD005036, 2004.

Lenot, X., Achard, V., and Poutier, L.: SIERRA: A new approach to atmospheric and topographic corrections for hyperspectral imagery, Remote Sensing of Environment, 113, 1664–1677, https://doi.org/10.1016/j.rse.2009.03.016, 2009.

Neumark, S.: Chapter 3 - Quartic equation, in: Solution of Cubic and Quartic Equations, edited by Neumark, S., pp. 12–24, Pergamon, https://doi.org/10.1016/B978-0-08-011220-6.50006-8, 1965.

Olson, M., Rupper, S., and Shean, D. E.: Terrain Induced Biases in Clear-Sky Shortwave Radiation Due to Digital Elevation Model Resolution for Glaciers in Complex Terrain, Frontiers in Earth Science, 7, https://doi.org/10.3389/feart.2019.00216, 2019.

Tuzet, F., Dumont, M., Picard, G., Lamare, M., Voisin, D., Nabat, P., Lafaysse, M., Larue, F., Revuelto, J., and Arnaud, L.: Quantification of the radiative impact of light-absorbing particles during two contrasted snow seasons at Col du Lautaret (2058 m a.s.l., French Alps), The Cryosphere, 14, 4553–4579, https://doi.org/10.5194/tc-14-4553-2020, 2020.

Extra figures:

[Figure]

New Figure 3: *Location of the study area, around the Col du Lautaret alpine site. The blue rectangle in (a) represents the* extension of the study area*, shown in (b). The domain is generated from the RGE ALTI®Version 2.0 Digital Elevation Model (DEM) provided by IGN France at a spatial resolution of 5m, and resampled to 10m for this study.* Slope angle is represented by the intensity of the color.

[Figure]

New Figure 5

[Figure]

New Figure 7

[Figure]

New Figure 8*: Comparison of the spatial variations of surface temperature between the simulations and the satellite observations for each date, computed considering the whole domain. On the left, mean bias and standard deviation of the differences. On the right, the RMSE and the correlation coefficient r.*

[Figure]

New Figure 11

[Figure]

New Figure 12